# Unravelling the role of redox active sites in nitrogen doped cerium oxide for associative ammonia decomposition

Dongpei Ye [1,6], Mingyu Luo[1,2,6], Xiaowei Liu[3], Christopher Foo[4], Mengqi Duan[1], Xuelei Pan [1], Jiasi Li[1], Simson Wu[5], Wei Liu [3], Michail Stamatakis [2] ✉, Yiyang Li [1] ✉ & Shik Chi Edman Tsang[1,7]

The catalytic decomposition of ammonia under mild conditions is a promising route for green hydrogen production. However, conventional dissociative ammonia decomposition pathways over metal sites are suffering from the Brønsted−Evans−Polanyi (BEP) constraint which establishes an inverse correlation between atomic N binding energy and the N-H bond dissociation energy. Herein, we report a ruthenium-supported nitrogen-doped cerium oxide (Ru/N-CeO$_2$) catalyst that breaks this limitation and exhibits significantly enhanced catalytic activity compared to its undoped counterpart. Furthermore, we reveal that N dopants can act as independent active sites, enabling an associative mechanism distinct from the conventional Ru-driven pathway. Comprehensive isotopic labelling experiments together with computational techniques elucidate the reaction mechanism over the N site and reveal a distinct correlation between the location of the active site and catalytic activity. The proximal N site exhibits the highest activity, challenging the conventional view that activity is dominated by metal−support interfacial sites. While N doping is a commonly used approach for surface modification, our findings show that it can also alter the reaction mechanism by introducing new active sites. These insights offer valuable guidance for the rational design of catalytic supports in ammonia decomposition and open new directions for catalytic systems limited by scaling relationships in heterogenous catalysis.

The thermo-catalytic decomposition of ammonia (NH$_3$) has garnered significant interest from an industrial perspective as a carbon-free route to hydrogen production. This reaction is highly attractive not only because it delivers hydrogen (H$_2$), with nitrogen (N$_2$) as its only side product, aligning with the principles of green energy, but also due to ammonia's high hydrogen-storage capacity and ease of liquefaction under mild conditions (i.e., ~8 bar at room temperature)[1]. However,

NH$_3$ decomposition is an endothermic and energy-intensive process, requiring efficient catalysts to lower activation barriers. Ruthenium (Ru) is found to be the state-of-the-art metal catalyst with outstanding catalytic activity, due to Ru's low activation energy for NH$_3$ dissociation, as well as high stability[2]. In conventional Ru-based catalysts for NH$_3$ decomposition, the reaction is widely accepted to proceed via a dissociative pathway, with possible rate-determining steps (RDS) being

[1]Wolfson Catalysis Centre, Department of Chemistry, University of Oxford, Oxford, UK. [2]Inorganic Chemistry Laboratory, Department of Chemistry, University of Oxford, Oxford, UK. [3]State Key Laboratory of Catalysis, Dalian Institute of Chemical Physics, Chinese Academy of Sciences, Dalian, China. [4]Diamond Light Source Ltd, Harwell Science and Innovation Campus, Didcot, UK. [5]Oxford Green Innotech Limited, 9400 Garsington Road, Oxford Business Park, Oxford, UK. [6]These authors contributed equally: Dongpei Ye, Mingyu Luo. [7]Deceased: Shik Chi Edman Tsang ✉e-mail: michail.stamatakis@chem.ox.ac.uk; yiyang.li@chem.ox.ac.uk

the dehydrogenation of adsorbed $NH_x$ species via N-H bond scission events or $N_2$ associative desorption[3].

The barriers of these processes are constrained by the Brønsted-Evans-Polanyi (BEP) scaling relationship, which dictates an inverse correlation between the atomic N binding energy and the N-H bond activation barrier[4–6]. During the reaction, stronger interaction between the atomic N from the reactant and a metal M active site would facilitate N-H bond dissociation due to the stronger back-donation of electrons from the metal sites to the anti-bonding orbital of the N-H bond. However, this would also inhibit the final $N_2$ desorption barrier from the metal surface, as the M-N interaction (quantified by the N binding energy) is too strong[7]. This scaling constraint fundamentally limits further performance improvements of Ru-based catalysts under mild conditions.

Biological systems in nature, such as nitrogenase enzymes, provide inspiration to overcome such limitations. Rather than cleaving the N ≡ N triple bond via direct dissociation, these enzymes facilitate stepwise hydrogenation at redox-active sites, effectively bypassing the BEP constraints and achieving high conversions at ambient conditions[8]. Inspired by this principle, we propose that nitrogen dopants in a redox-active support can mimic enzymatic behaviour – introducing alternative, so called "associative" reaction pathways and unlocking new design opportunities for heterogeneous catalysis. This strategy provides a route to relax or overcome BEP constraints in heterogeneous catalysis and facilitate ammonia decomposition under milder conditions.

In this context, nitrogen-doped cerium oxide ($N-CeO_2$) emerges as a promising support. $CeO_2$ is well known for its outstanding redox properties due to the facile interconversion between $Ce^{3+}$ and $Ce^{4+}$, and it can incorporate nitrogen species into its lattice under appropriate conditions, forming N-substituted oxygen sites that mimic the behaviour of metal nitrides while providing superior air stability[9–11]. Typically, nitrogen doping is an approach for modifying the support surface to offer higher basicity, stronger metal-support interaction, and better metal dispersion[12–14]. However, the role of nitrogen species in $CeO_2$ is inherently complex due to the presence of mobile oxygen vacancies and has rarely been investigated, especially in terms of the potential of such species to act as independent active sites for ammonia decomposition.

In this study, we present a catalyst consisting of ruthenium nanoparticles (NPs) supported on nitrogen-doped cerium oxide (Ru/N-$CeO_2$) that demonstrates significantly enhanced $NH_3$ decomposition activity compared to its undoped Ru/$CeO_2$ counterpart. Through a combined isotopic pulse infrared spectroscopy technique (IsP-FTIR), together with advanced theoretical approaches, including density functional theory (DFT) and machine learning-based atomic cluster expansion (MACE), we provide strong evidence that nitrogen dopants can act as independent redox-active sites, enabling an associative reaction mechanism involving a key $^{14}N-^{15}N*$ intermediate. Crucially, we discover a spatial correlation of activity, where proximal N-dopants, rather than interfacial Ru-support sites, exhibit the highest catalytic activity. Additionally, surface characterisation further illustrates that the surface reconstructs upon N doping, leading to enhanced electron donation to the metal centre and facilitating dual-site reactivity. These insights collectively demonstrate that nitrogen dopants located on specific support sites can alter the catalytic mechanism, breaking traditional scaling constraints and offering a new strategy for efficient $NH_3$ decomposition under mild conditions. This work offers a new framework for designing heterogeneous catalysts for $NH_3$ activation, focusing particularly on the metal-support interface.

## Results
### Structural and surface analysis
$CeO_2$ was synthesised using a typical soft urea glass (SUG) method, while its nitrogen doped derivatives were obtained by subsequent annealing under $NH_3$ flow at various temperatures and durations (Fig. 1a). These derivatives are denoted as $N-CeO_2$ (T-t), where T and t represent temperature in °C and duration in h, respectively, e.g. Ru/N-$CeO_2$ (500-10) was annealed at 500 °C for 10 h. The structural properties of the material are comprehensively characterised using X-ray diffraction (XRD), scanning electron microscopy (SEM) and nitrogen Brunauer-Emmett-Teller (BET) surface area analysis (Figures S1 and S2, Table S1 and Supplementary discussion 1).

Elemental analysis and chemical properties were investigated via X-ray photoelectron spectroscopy (XPS). High resolution N $1s$ spectroscopy evidences the existence of N and indicates its successful doping into the cerium oxide material (Figure S3 and Supplementary discussion 2)[15]. The peak located at 399.2 eV in the spectrum indicates that N is doped into the structure[16]. Further quantification of the XPS spectra and Carbon Hydrogen Nitrogen Sulphur (CHNS) analysis reveal that the level of N dopants in the material is strongly related to the temperature. Table S1 thus presents a 'bell' shaped trend, where the highest amount of nitrogen is found for a sample annealed at 650 °C with an atomic nitrogen ratio of ca. 2.76%. This aligns with the optimal temperature used to achieve maximum uptake of N via ammonolysis on metal oxides[11,17,18]. Additionally, with increasing annealing temperature, the $Ce^{3+}/Ce^{4+}$ ratio increases (Figure S4). This phenomenon entails $O^{2-}$ donating electrons to $Ce^{4+}$ and leaving the lattice, thereby giving rise to oxygen vacancies ($O_v$). Higher ammonolysis temperature results in the formation of more surface oxygen vacancies/defects, which are stabilised by such $Ce^{3+}-O_v$ interaction[19,20].

The redox properties of the $CeO_2$ materials are analysed using $H_2$ temperature programmed reduction ($H_2$-TPR). Two distinct reduction peaks corresponding to surface oxygen removal (200–300 °C) and reduction of the outer layer of $CeO_2$ to $Ce_2O_3$ (350–500 °C) are observed (Figure S5)[21,22]. The reduction temperature directly reflects the involvement of surface-active oxygen species and is intricately linked to how easy oxygen vacancies form. The observed reduction peak shift suggests that nitrogen doping enhances the reducibility of neighbouring Ce ions and promotes greater mobility for oxygen vacancies[23,24]. Furthermore, the extent of this shift correlates well with the nitrogen content measured by XPS and CHNS analyses. While oxygen vacancies are not solely generated by heteroatom incorporation into the lattice, nitrogen dopants facilitate their formation through charge compensation mechanisms. This intrinsic modification of the redox properties plays a critical role in catalytic activity and is further explored upon Ru loading in the context of $NH_3$ decomposition.

### Evaluation of Catalytic Performance
As mentioned, Ru is the state-of-the-art metal for $NH_3$ decomposition. Thus, catalytic $NH_3$ decomposition was performed over Ru-loaded cerium oxide and nitrogen-doped cerium oxide support to evaluate the catalyst's performance. By screening the synthesis conditions, an optimal catalyst Ru/N-$CeO_2$ (650-6), yielding a conversion of ca. 99% at 500 °C and weight hourly space velocity (WHSV) of 30,000 ml $g_{cat}^{-1}$ $h^{-1}$ was obtained (Fig. 1b, c). The structural and chemical properties of the Ru-loaded catalysts were studied and the results illustrated that generally higher nitrogen concentration led to higher activity (Figure S6, Table S1 and Supplementary discussion 3). Nitrogen doping is commonly considered as an approach to promote catalytic activity of metal-based catalysts due to the electron donation effects[14,25]. The lone pairs of electrons of N atoms can increase the electron donating properties of the support surface, hence leading to increasing basicity of the support. Nitrogen doping also facilitates electron transfer from the support to Ru and increases the activity of Ru for $NH_3$ decomposition[25,26]. A series of catalysts with different Ru loadings and N contents was synthesised. The results showed that with an optimum N loading, a 2- to 3-fold reduction in the required Ru loading could be achieved (Figure S7 and supplementary discussion 4).

Further catalytic measurements and subsequent characterisations were carried out on the most promising sample, Ru/N-CeO$_2$ (650-6) (Ru content: 1.63 wt.%), which offers an optimal balance between high catalytic performance and relatively low Ru loading. The WHSV was varied and an almost equilibrium conversion of ca. 96% at 450 °C at a WHSV of 15,000 ml g$_{cat}^{-1}$ h$^{-1}$, which outperforms most of the state-of-the-art catalysts (Table 1 and S2), is achieved (Fig. 1d). It is as expected that NH$_3$ conversion decreases with the increase of WHSV owing to the shorter residence time for NH$_3$ absorbed on active sites. To access long-term stability, the Ru/N-CeO$_2$ (650-6) sample was placed under continuous flow of NH$_3$ at 450 °C and 30,000 ml g$_{cat}^{-1}$ h$^{-1}$ for 70 h. As illustrated in Fig. 1e, no significant drop in the conversion is observed. Moreover, XRD and SEM have been carried out over the catalyst after the catalysis (Figure S8 and S9). The results demonstrate strong structural stability of N-CeO$_2$ that holds throughout the catalysis testing, thereby highlighting its potential for industrial applications in NH$_3$ decomposition.

Kinetic studies were performed in the kinetically limited regime of the Ru/CeO$_2$ and Ru/N-CeO$_2$ (650-6) catalysts and the results were compared. The apparent activation energy ($E_a$) of Ru/CeO$_2$ is found to be 90.5 ± 1.9 kJ mol$^{-1}$, which lies in the range of 80–115 kJ mol$^{-1}$ for traditional Ru catalysts following the dissociative pathway (Fig. 1f)[27,28]. However, with N doping, the $E_a$ of Ru/N-CeO$_2$ (650-6) drops significantly to 65.2 ± 2.6 kJ mol$^{-1}$. The activation energy reflects how facile is the overall reaction's rate determining step (RDS), which is in general considered to be N-H bond scission on metal sites for Ru based systems[29]. Thus, the low $E_a$ value for the Ru/N-CeO$_2$ (650-6) which lies out of the range of typical dissociative NH$_3$ decomposition mechanism over Ru based catalyst would suggest a shift in the overall reaction pathway.

### Effect of nitrogen doping on Ru

The catalytic activities of plain support CeO$_2$ and N-CeO$_2$ as well as their Ru-impregnated catalyst counterparts have been studied and compared (Figure S10). Not surprisingly, the plain supports showed

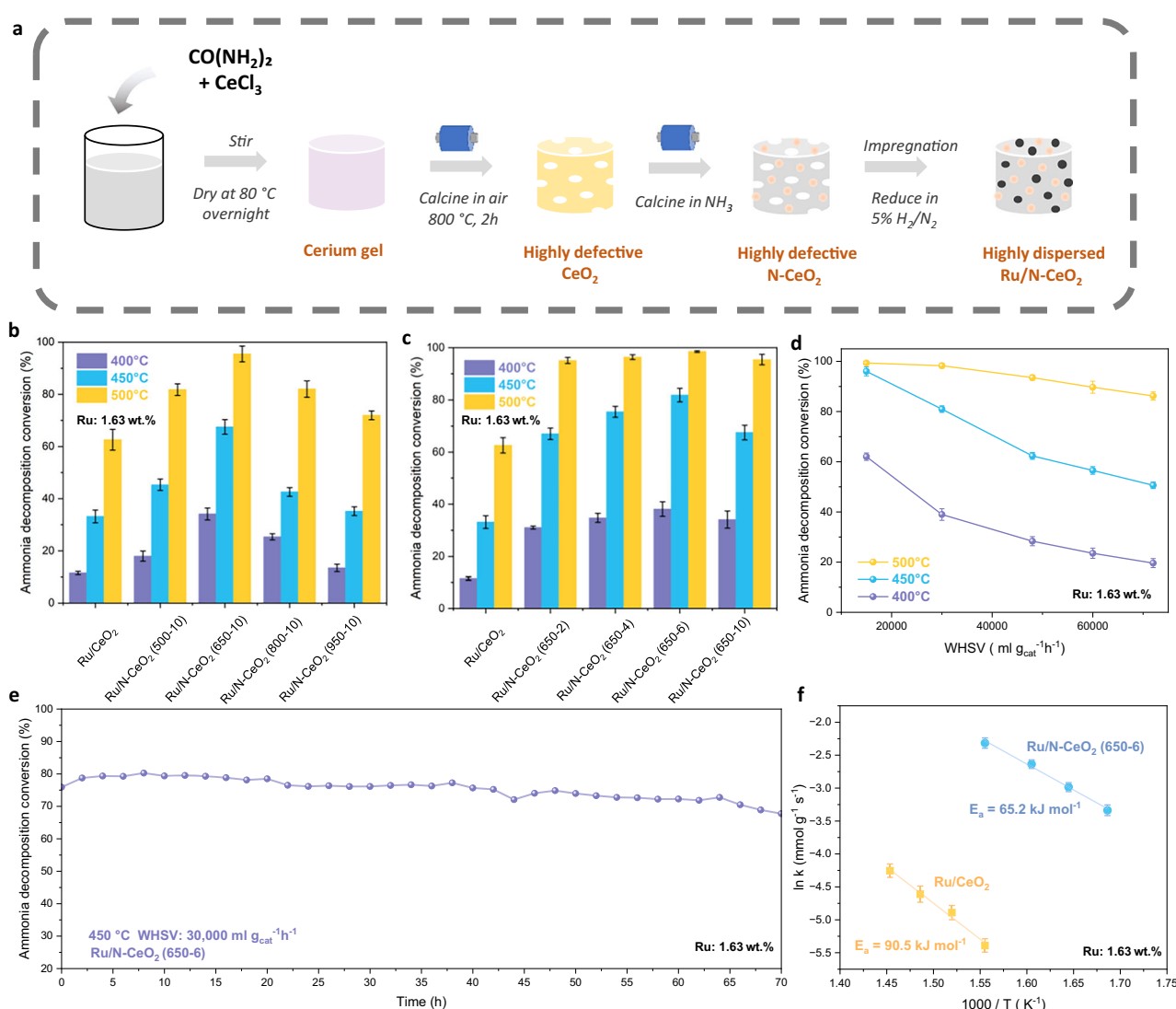

**Fig. 1 | Synthetic procedure and catalytic analysis of NH$_3$ decomposition on the ruthenium NPs supported on nitrogen-doped cerium oxide catalysts.**
**a** Schematic illustration of the synthetic process of N doped cerium oxide.
**b**, **c** Performance of NH$_3$ decomposition reaction at WHSV = 30,000 ml g$_{cat}^{-1}$ h$^{-1}$ over Ru on CeO$_2$ and N-CeO$_2$ treated in NH$_3$ at different temperature (**b**) and duration (**c**). **d** Catalytic conversion of NH$_3$ decomposition on Ru/N-CeO$_2$ (650-6) catalyst at different WHSV. **e** Stability test of Ru/N-CeO$_2$ (650-6) catalyst under NH$_3$ gas flow for 70 h. **f** Arrhenius plot of Ru/CeO$_2$ and Ru/N-CeO$_2$ (650-6). Error bars represent the mean ± standard deviation from 3 independent measurement. Each measurement was conducted under identical conditions using a freshly prepared batch of catalyst sample.

**Table 1 | Catalytic activity comparison with Ru-based catalysts for ammonia decomposition**

| Catalyst | Metal content /wt. % | Reaction Temp. / °C | WHSV /ml $g_{cat}^{-1}$ $h^{-1}$ | $NH_3$ Conv. /% | Apparent $H_2$ formation rate /mmol $g_{cat}^{-1}$ $min^{-1}$ | Reference |
|---|---|---|---|---|---|---|
| Ru/CNTs | 5.0 | 450 | 30,000 | 43.7 | 14.6 | 66 |
| Ru/CNFs | 3.2 | 500 | 6500 | 99.0 | 6.2 | 67 |
| Ru/La$_{0.33}$ Ce$_{0.67}$ | 1.8 | 450 | 6000 | 99.9 | 6.7 | 31 |
| Ru/CeO$_2$ | 1.0 | 350 | 22,000 | ca. 32.0 | 8.1 | 68 |
| Ru/c-MgO | 4.7 | 450 | 30,000 | 80.6 | 26.5 | 69 |
| Ru/MgO (111) | 3.4 | 450 | 30,000 | 99.9 | 32.4 | 46 |
| Ru/C12A7:e⁻ | 2.2 | 450 | 15,000 | 99.9 | 16.7 | 70 |
| Ru/Ce$_5$/MgAl(600) | 2.0 | 465 | 30,000 | 50.0 | 16.8 | 71 |
| RuLaCs/Al$_2$O$_3$ | 1.0 | 450 | 5000 | 99.0 | 5.2 | 72 |
| Ru/CeO$_2$NR-v | 0.5 | 450 | 15,000 | ca. 80.0 | 13.3 | 73 |
| Ru/CeO$_2$ | 1.6[a] | 450 | 30,000 | 43.2 | 14.5 | This work |
| Ru/N-CeO$_2$ (650-6) | 1.6[a] | 450 | 15,000 | 96.0 | 16.5 | This work |
| Ru/N-CeO$_2$ (650-6) | 1.6[a] | 450 | 30,000 | 79.5 | 26.6 | This work |
| Ru/N-CeO$_2$ (650-6) | 1.6[a] | 450 | 72,000 | 50.6 | 40.6 | This work |

[a]Ru content of this work obtained using ICP-MS

low activities, and even though the N-doped support exhibits a slightly higher activity than plain CeO$_2$, it is clear that the presence of Ru is essential for high catalytic performance. However, the origin of the observed activity remains to be clarified; specifically, the key question is whether the activity arises solely from Ru B$_5$ sites operating via the conventional dissociative mechanism, or whether synergistic interactions between Ru and N dopants also contribute through an associative pathway. H$_2$-TPR on freshly prepared Ru supported on CeO$_2$ and N-CeO$_2$ materials has been carried out. It exhibits a clear peak in the range of 90–150 °C, attributed to the reduction of oxidised Ru species (Ru$^{n+}$) to metallic Ru$^0$ (Figure S11)[30]. This peak shifts remarkably from 95 up to 144 °C for Ru on various nitrogen-doped CeO$_2$. During wetness impregnation, the lone pairs on surface-exposed nitrogen sites interact with the positively charged Ru species, resulting in stronger Ru metal support interaction (MSI). This enhanced MSI manifests as a shift of the Ru reduction peak to higher temperature and gives rise to the linear correlation observed between the surface nitrogen content and the reduction temperature of the Ru species (Fig. 2a).

The Ru-containing catalysts were analysed using X-ray absorption spectroscopy (XAS) in three different stages: (i) as synthesised after wetness impregnation (AS), (ii) after reduction (RED), and (iii) after catalysis (AC). As shown in Fig. 2b, c, Ru on both Ru/N-CeO$_2$ (650-6) AS and Ru/CeO$_2$ AS catalysts appears predominantly as RuO$_2$. However, a shift to lower R value is noted for the N doped support. Wavelet transformation analysis (Fig. 2d, e) further substantiates this difference, suggesting the distinct existence of Ru-N interaction on Ru/N-CeO$_2$ (650-6) AS catalyst. Notably, Ru−N bond exhibits a shorter bond distance compared to Ru−O, consistent with the higher bond strength of Ru−N[31]. Moreover, the Ru-N coordination path appears at a lower k-space value compared to Ru-O, further validating the interaction between Ru species and nitrogen sites on the support[32,33]. These observations suggest that during impregnation, Ru species interact with surface nitrogen dopants, providing a chelating and anchoring effect. This conclusion is supported by the H$_2$-TPR data, where stronger MSI is observed for higher nitrogen dopant content. After reduction, a significant increase in the metallic Ru-Ru bonding peak located at 2.67 Å is observed, accompanied by the disappearance of the Ru-O/ Ru-N bond (Fig. 2c and S12). This is in line with the Ru 3p XPS spectrum that shows a predominant Ru metallic phase (Fig. 2f)[34]. Nevertheless, both analytic techniques indicate that the reduced samples still exhibit a higher average oxidation state compared to metallic Ru. The presence of O$_v$ and nitrogen dopants on the support would result in a

partial charge crossing the interphase between the Ru NPs and the cerium oxide support. This interfacial electronic effect is likely to contribute to a change in the oxidation state of Ru species, which results in the peak shift observed in the spectrum[35,36]. On another note, the XPS 3p$_{3/2}$ spectrum of Ru$^0$ shows a peak position deviation from 461.4 eV for Ru/N-CeO$_2$ (650-6) compared to 462.0 eV for the undoped counterpart. This suggests that for fully reduced samples, the electron density on Ru is still slightly higher for N-CeO$_2$ (650-6) compared to pristine support, which agrees with the electron transfer effect from N lone pair to the Ru metal as discussed above. However, the low nitrogen content of the sample renders such effects subtle and makes them difficult to identify in the XAS spectra. Moreover, the Ru coordination number on the N-doped support is smaller compared to the pristine counterpart. This re-emphasises the anchoring effect of N towards Ru stabilisation. This effect becomes more prominent after catalysis, where the Ru coordination number exhibited a larger increase for Ru/CeO$_2$ (Figures S13, 14; Table S3) due to particle sintering, reinforcing that N incorporation can help stabilise the Ru and inhibit further aggregation[37,38]. These findings further emphasise the importance of surface nitrogen configuration, as nitrogen dopants help anchor Ru NPs, limiting their mobility and preventing sintering.

To better visualise and analyse the size and morphology of the catalyst, transmission electron microscopy (TEM) was employed. As illustrated in Fig. 2g and Figure S15, Ru NPs exhibit a relatively broad size distribution (from < 1 to 5 nm), rather than a single uniform size. Ru NPs are well dispersed across the surface of N-CeO$_2$ (650-6) with an average size of 2.1 ± 0.9 nm. The corresponding energy-dispersive X-ray spectroscopy (EDS) mapping further demonstrates the homogeneous distributions of Ce, O, N and Ru on the catalyst (Figure S16). The Ru NPs on pristine CeO$_2$ shows a slightly higher average particle size of 2.9 ± 0.7 nm (Fig. 2h) owning to the anchoring effect of N dopants as mentioned above. Such a Ru-N interaction would stabilise the Ru NPs during synthesis and leading to smaller particle size and better dispersion.

During initial calcination of the cerium gel mixture, evaporation and combustion of urea leave behind surface pits and voids (Fig. 2i, j). Structural defects observed on the N-CeO$_2$ (650-6) support matches in size with the observed Ru NPs located on the support's (110) surface (Fig. 2k, l), indicating that these structural depressions facilitate the formation of finely dispersed NPs by serving as additional anchoring sites. Furthermore, the removal of urea may also lead to defects exposing reactive higher-index crystal planes, at which N is

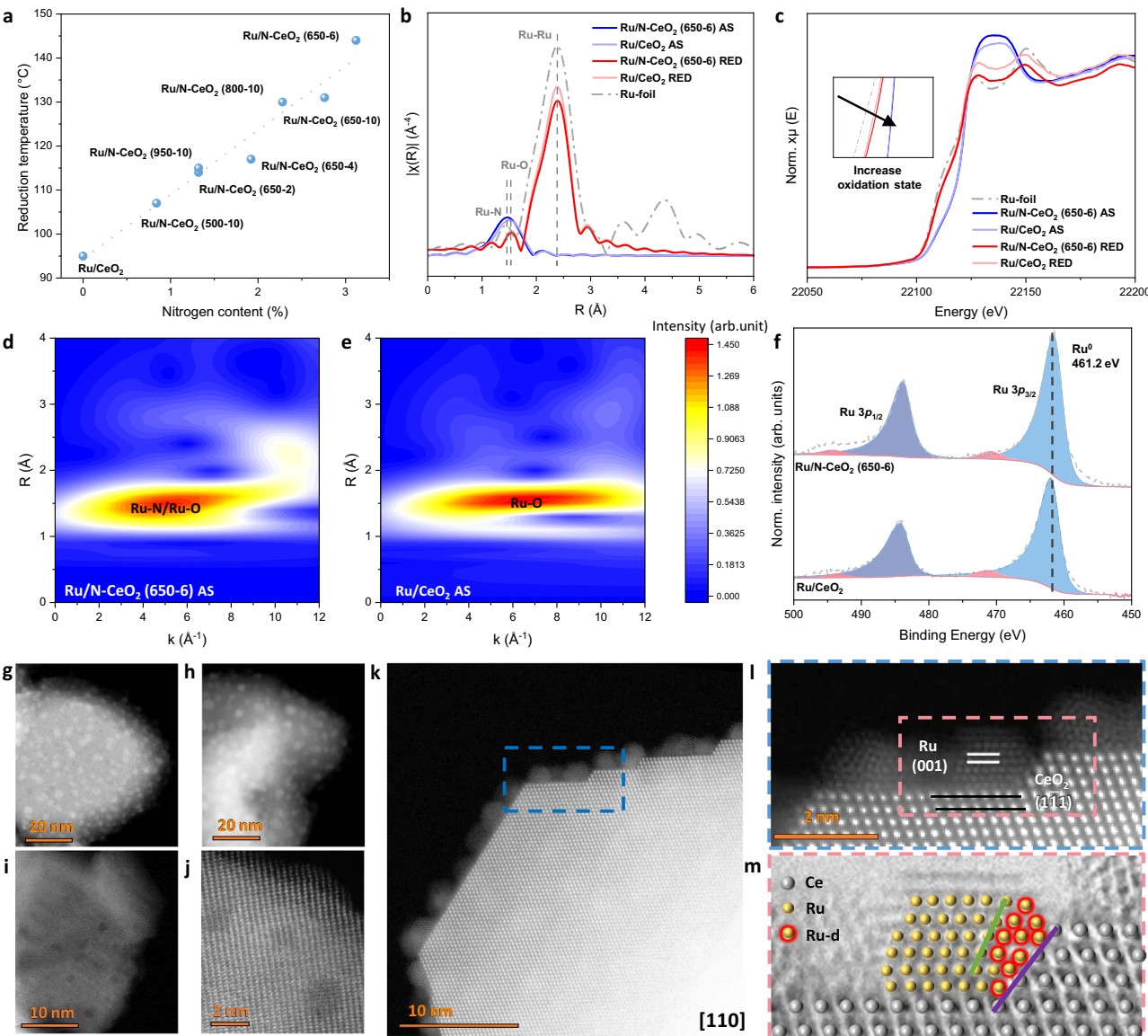

**Fig. 2 | Chemical properties of Ru loaded CeO₂ and N doped CeO₂. a** Linear correlation between the N dopant concentration and the Ru reduction temperature from H₂-TPR. X-Ray absorption spectroscopy on the Ru $K$-edge of ruthenium NPs supported on nitrogen-doped cerium oxide catalysts. **b** The plot of $k^3$-weighted Fourier transforms of the Ru K-edge EXAFS spectra. **c** XANES spectrum. **d, e** Wavelet transformation plots for the $k^3$-weighted EXAFS signal of Ru/N-CeO₂ (650-6) AS (**d**), intensity maximum at (5.01 Å⁻¹, 1.49 Å) and Ru/CeO₂ AS (**e**), intensity maximum at (6.64 Å⁻¹, 1.57 Å). **f** Ru $3p$ XPS spectra of the ruthenium NPs supported on cerium oxide and nitrogen-doped cerium oxide catalysts. **g, h** High-angle annular dark-field scanning transmission electron microscopy (HAADF-STEM) of Ru/N-CeO₂ (650-6) (**g**) and Ru/CeO₂ (**h**). **i, j** HAADF images showing the plain N-CeO₂ (650-6) support without Ru impregnation. **k, l** High resolution HAADF images showing Ru NPs located on the (110) plane of N-CeO₂ (650-6) support. **m** Annular bright-field scanning transmission electron microscopy (ABF-STEM) of Ru NPs supporting on N-CeO₂ (650-6), illustrating the distortion at the Ru (011) (green line) and support's (001) (purple line) surface, the distorted Ru atoms are highlighted with red outer shells.

preferentially doped[39]. Upon Ru impregnation and subsequent reduction, it will lead to local structural distortion as a result of oxygen vacancy formation and N atom doping (Fig. 2m)[40,41]. Such an effect can enhance the MSI by modifying the local coordination environment and electronic structure.

### In-situ analysis and mechanistic understanding

*Operando* NH₃ FTIR measurements were carried out and analysed to understand the reaction mechanism for NH₃ decomposition (Figures S17 and S18, and Supplementary discussion 5). As temperature is gradually increased, a broad and weak peak representing N = N bond at 1430 cm⁻¹ shows up (Figure S17) for N-doped catalysts[42,43]. Additionally, peaks at 2163, 1970 cm⁻¹ (attributed to N≡N stretching of N₂ molecules formed by inducing NH₃ decomposition) and 2060 cm⁻¹ (ascribed to a key *N₂H intermediate species that is being detected on the surface of the support) are observed for plain N-CeO₂ (650-6) support at elevated temperature (Fig. 3a)[42,44,45]. These features are most significantly observed on plain N-CeO₂ (650-6) support, which indicates that the redox active N dopants can activate catalytic reactions by inducing a N-N interaction with the incoming NH₃ molecule (Supplementary discussion 6). In this context, NH₃ is directly adsorbed on the N sites of the catalyst support via an associative manner. This Mars-van Krevelen (MvK) like pathway further suggests that NH₃, instead of dissociating completely all the way to N, would undergo stepwise dehydrogenation accompanied by N-N bond formation via a *N=NH_x intermediate[7].

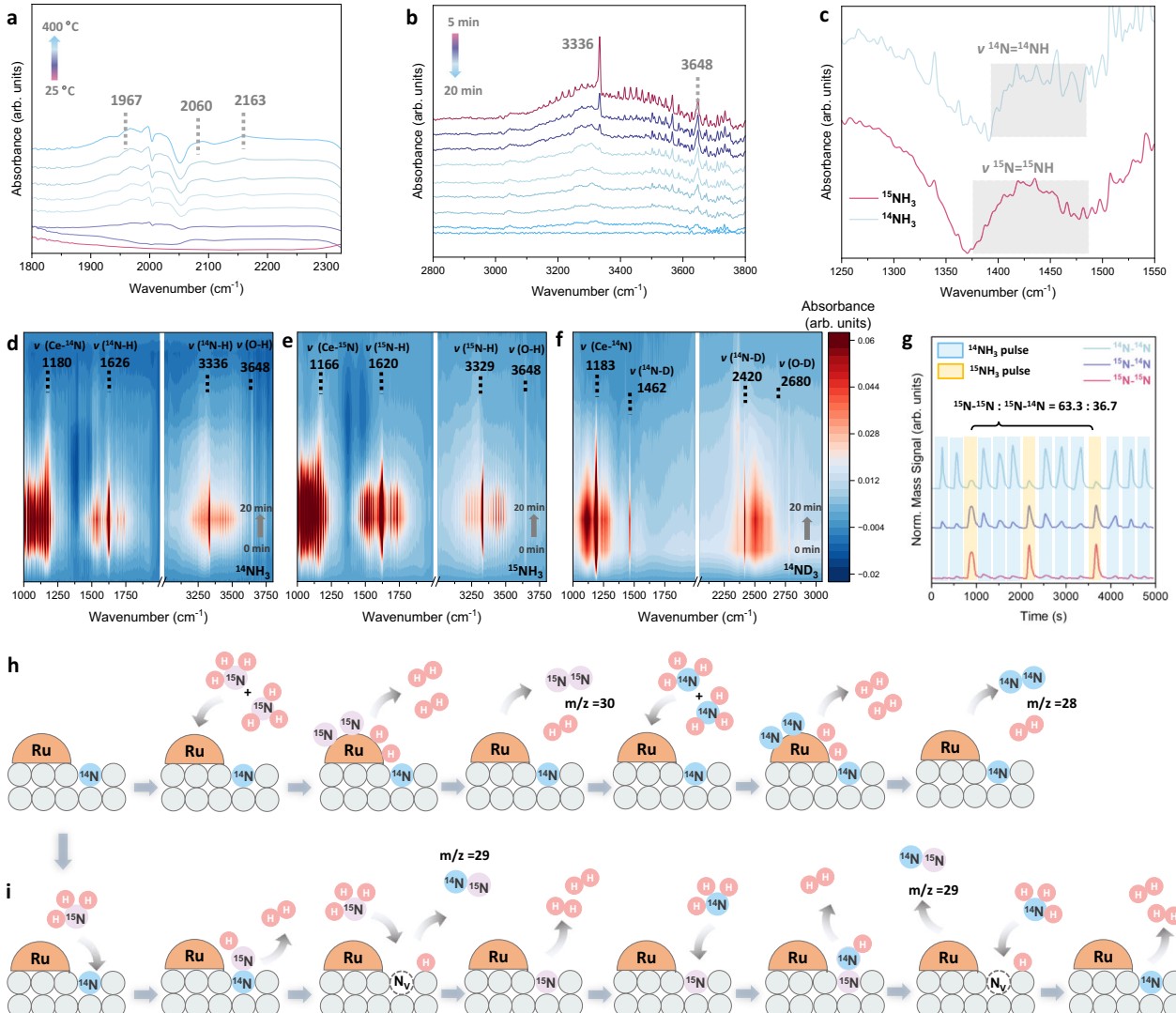

**Fig. 3 | *Operando* FTIR, IsP-FTIR and isotopic labelling experiments for mechanism study. a** *Operando* $^{14}NH_3$ FTIR spectrum of N-CeO$_2$ (650-6) from 25 to 400 °C, under 10 ml min$^{-1}$ 10% NH$_3$/He flow **b** Time resolved $^{14}NH_3$ pulse experiment on Ru/N-CeO$_2$ (650-6) at 400 °C and 10 ml min$^{-1}$ Ar flush, pulse contains 0.6 bar of reactant gas. **c** Comparison between the $^{15}NH_3$ and $^{14}NH_3$ isotopic pulse FTIR (IsP-FTIR) experiments on Ru/N-CeO$_2$ (650-6) at 400 °C showing the N=NH peak shift (each spectrum taken 5 min after the pulse, each isotopic pulse consists of 0.6 bar of reactant gas). **d**–**f** 2D counter plot of the IsP-FTIR on Ru/N-CeO$_2$ (650-6) at

400 °C, 10 ml min$^{-1}$ flow of Ar, each pulse contains 0.6 bar of (**d**). $^{14}NH_3$; (**e**) $^{15}NH_3$ and (**f**) $^{14}ND_3$. **g** $^{15}NH_3$/$^{14}NH_3$ pulse experiment on MS. Operated at 450 °C, on Ru/N-CeO$_2$ (650-6) with 10 ml min$^{-1}$ flow of Ar, each pulse contains 0.6 bar of either $^{15}NH_3$ or $^{14}NH_3$. **h** Dissociative NH$_3$ decomposition pathway that takes place over Ru sites, giving rise to m/z = 28 and m/z = 30 signal. **i** Proposed associative pathway following the Mars-van Krevelen mechanism on the active nitrogen sites, giving rise to the m/z = 29 signal (the m/z = 28 or 30 signals are also possible for this pathway, depending on the state of the surface and what is being pulsed in the reactor).

Further isotopic $^{15}NH_3$/$^{14}ND_3$/$^{14}NH_3$ pulse IsP-FTIR experiments were carried out to evaluate the catalytic activity and reaction intermediates in more detail. This time-resolved approach helps capture short-lived intermediates by introducing reactants in short, controlled bursts, allowing transient species to be observed before they are fully converted to products. The same four characteristic NH$_3$ peaks are observed during the $^{14}NH_3$ pulse over the Ru/N-CeO$_2$ (650-6) catalyst (Figure S19a). These peaks decrease in intensity over time, consistent with the progressive decomposition of NH$_3$ into nitrogen and hydrogen gas. A particularly notable observation is the emergence of a broad O-H vibrational band around 3500 cm$^{-1}$, along with a sharp, isolated O-H peak at 3648 cm$^{-1}$, attributed to surface hydroxyl groups (Fig. 3b). Both peaks diminish during Ar purging, indicating their transient nature under reaction conditions. Concurrently, a distinct band at 1918 cm$^{-1}$, assigned to Ru-H stretching, also diminishes over time (Figure S19b)[46]. The synchronous emergence of O-H and Ru-H peaks during NH$_3$ decomposition strongly suggests that the surface

hydroxyls originate from the dissociation of NH$_3$. The hydrogen atoms generated during decomposition appear to migrate from Ru active sites towards the surface, reacting with surface oxygen to form O-H groups. This hydrogen migration directly indicates that the Ru/N-CeO$_2$ (650-6) catalyst exhibits a notable hydrogen spill-over effect. $^{14}ND_3$ pulse experiment further corroborated the idea, where the broad O-H and isolated O-H peaks shift to ~2500 and 2680 cm$^{-1}$, respectively (Figure S19c). These shifts are consistent with expectations from the harmonic oscillator model for isotopic substitution (3500 cm$^{-1}$ × ($\mu_{OH}$/$\mu_{OD}$)$^{0.5}$ = 2500 cm$^{-1}$ and 3648 cm$^{-1}$ × ($\mu_{OH}$/$\mu_{OD}$)$^{0.5}$ = 2630 cm$^{-1}$). Corresponding shifts in N-H and N-D bond frequencies were also noted. Comparative analysis of spectra from $^{14}NH_3$ and $^{15}NH_3$ pulses also reveals a red shift in the N=N stretching region, with the peak moving from 1430 to 1405 cm$^{-1}$ (Fig. 3c). This shift corresponds to the formation of $^{15}N$=$^{15}N$ (1382 cm$^{-1}$) and $^{15}N$=$^{14}N$ (1406 cm$^{-1}$) species, confirming the nitrogen interaction between NH$_3$-derived nitrogen and lattice nitrogen. Additionally, the Ce-NH$_x$ adsorption peak shifts from 1180 to

1166 cm$^{-1}$ (Fig. 3d–f and Figure S19d)[47,48]. This red shift lends further support to an associative mechanism, where NH$_3$-derived intermediates engage directly with surface sites on the catalyst and probe further NH$_3$ decomposition.

To further verify the reaction mechanism, $^{15}$NH$_3$/$^{14}$NH$_3$ pulse experiments on a mass spectrometer (MS) were carried out. When $^{14}$NH$_3$ pulses were initially pumped into the reactor (the first two pumps), only peaks at m/z = 28 which represents $^{14}$N-$^{14}$N arise (Fig. 3g). Relatively small changes were noted for signal m/z = 29 ($^{14}$N-$^{15}$N) and m/z = 30 ($^{15}$N-$^{15}$N), which were considered as a change in baseline due to pulsing extra pressure to the system. This is as expected, since no $^{15}$N is yet present in the system. However, when $^{15}$NH$_3$ is pulsed into the system, the signal intensities of m/z = 29 ($^{15}$N-$^{14}$N) and m/z = 30 ($^{15}$N-$^{15}$N) both increases. Under the dissociative mechanism, an N$_2$ molecule is formed by bare N atoms originating from the NH$_3$ reactants, suggesting an expected rise in only the $^{15}$N-$^{15}$N peak. Therefore, a significant increase in the m/z = 29 peak clearly indicates processes taking place at the surface of N-doped cerium oxide. In particular, $^{15}$NH$_3$ bonds with the lattice $^{14}$N from the support, eventually leading to the formation of $^{15}$N-$^{14}$N, in line with the MvK route proposed in the discussion above. NH$_3$-TPD results show that NH$_3$ readily desorbs from the catalyst at a temperature lower than 300 °C (Figure S20). Furthermore, the XAS spectrum of the K-edge of Ru on Ru/CeO$_2$ and Ru/N-CeO$_2$ (650-6) before and after catalysis indicates no change in the oxidation state of the Ru NPs, thus, no nitridation taking place (Figures S13, 14). These two measurements reveal that by flushing argon after each pulse, no $^{15}$NH$_3$ or $^{14}$NH$_3$ would remain in the system, further corroborating that the observed $^{15}$N-$^{14}$N originates from reactions between the incoming $^{15}$NH$_3$ and lattice $^{14}$N. More interestingly, after pulsing $^{15}$NH$_3$, the $^{14}$NH$_3$ pulse right after it would also give rise to a m/z = 29 peak signal. This is because $^{15}$NH$_3$ would drag the lattice $^{14}$N out of the support, leading to the formation of vacancy sites (V$_N$) thereon. Under the $^{15}$NH$_3$ pulse, this V$_N$ is filled by $^{15}$N, which results in $^{14}$N and $^{15}$N exchange in the lattice of the catalyst. Thus, the next $^{14}$NH$_3$ pulse would also result in the formation of $^{15}$N-$^{14}$N, but this time, it is the $^{14}$N from $^{14}$NH$_3$ which interacts with the lattice $^{15}$N. As more $^{14}$NH$_3$ is pulsed, a decrease in the m/z = 29 peak signal is observed, owing to less lattice $^{15}$N remaining in the support. However, it is worth mentioning that the observed rise in the m/z = 30 peak (under $^{15}$NH$_3$ pulse) indicates that Ru still serves as an active site and the dissociative mechanism holds throughout the reaction (Fig. 3h–i). By quantifying the area of the peaks corresponding to $^{14}$N-$^{15}$N and $^{15}$N-$^{15}$N during each $^{15}$NH$_3$ pulse, it is revealed that during the catalytic reaction, approximately 36.7 ± 1.5% of the products arises from the associative pathway, while 63.3 ± 1.5% originate from dissociative mechanisms. The same experiment was carried out over Ru/CeO$_2$ under the same condition, but no significant rise of m/z = 29 peak was observed (Figure S21). This indicates that introducing N dopants to the support results in significantly higher contribution from the associative pathway, highlighting the promoting effect of N in the NH$_3$ decomposition catalysis. To sum up, this $^{15}$NH$_3$/$^{14}$NH$_3$ pulse experiment provides strong evidence that the N-doped cerium oxide catalyst carries out NH$_3$ decomposition not only by the traditional dissociative pathway, but also by the novel associative mechanism.

## Theoretical insights into the role of surface nitrogen dopants

To gain deeper mechanistic insights into the associative mechanism observed from the isotopic experiments, theoretical investigations were performed aiming to delineate the corresponding elementary steps and energetic landscape (Figures S20 and S21). Using the MACE-MP-0 interatomic potential, the role of nitrogen sites was evaluated along the MvK pathway as shown in Fig. 4a. In this MvK cycle, the doped lattice nitrogen acts as a redox-active reservoir. The cycle starts from a fully occupied N$_{doped}$ site. Successive dehydrogenation of NH$_3$ produces surface NH$_x$ intermediates, which couple with the lattice N$_{doped}$ to eventually form N$_2$. These steps progressively oxidise the lattice nitrogen N$^{3-}$ to N$^0$ and generate a nitrogen vacancy (V$_N$). The N$_2$ molecule then desorbs from the Ru/N-CeO$_2$ surface with an associated thermodynamic cost (NN$^*$ to N$_2$). Subsequently, nitrogen from incoming NH$_3$ refills V$_N$ via further dehydrogenation and N insertion, restoring the N$_{doped}$ site (NH$_3$$^*$ to slab). Together, N$_2$ desorption and V$_N$ refilling complete the MvK cycle and establish the lattice N sites as genuinely redox-active. According to the presented results, the N active site over pure N-CeO$_2$ support without Ru loading shows a relatively high thermodynamic barrier of 2.04 eV due to the NH$_3$$^*$ → NH$_2$$^*$ step. Upon Ru loading, this step is markedly facilitated, indicating the key role of Ru metal for N-H dissociation. Additionally, by evaluating the reaction pathways over Ru/N-CeO$_2$ models with different N to Ru distances, a correlation between the thermodynamic barrier and the spatial distance between the active site and Ru is revealed, with the N site proximal to the Ru nanoparticle exhibiting the lowest barrier of 1.45 eV (1.60 eV observed for interfacial N site and 1.88 eV for the distal N site). Furthermore, there is a shift in the RDS along different N active sites; in particular, the RDS shifts from the final N-H cracking step on the vacancy for the proximal N and distal sites, to N$_2$ desorption for the interfacial N site. These results indicate that while the Ru nanoparticle greatly promotes the N-H bond dissociation, too small a distance between the N site and Ru nanoparticle, which may correspond to excessively strong metal-support interaction, can impose a kinetic penalty and hinder the final N$_2$ desorption, since the latter process occurring from the interfacial N site entails a large barrier of 1.60 eV. On the other hand, the proximal N site, which keeps a moderate distance to the Ru nanoparticle, strikes a balance between these two factors and therefore exhibits the best activity. As shown in Figure S22, the calculated RDS barrier exhibits a non-monotonic dependence on the Ru-N$_{doped}$ distance: it drops from the interfacial (Ru-N bond distance 1.7 Å) to the proximal N site (Ru-N distance 4.1 Å) but increases again for the distal and Ru-free CeO$_2$ cases. This non-monotonic trend identifies the proximal N configuration as the optimum, giving the lowest RDS barrier and thus the highest predicted activity. Notably, the energy barrier for proximal N migration onto adjacent Ru nanoparticles is prohibitively high (1.92 eV, Figure S23), effectively suppressing dynamic coupling between the two distinct active sites – the Ru nanoparticle and proximal N. This substantial energy penalty confines their catalytic roles to independent pathways, further corroborating that the m/z = 29 signal in our isotopic pulse experiments originates exclusively from Mars-van Krevelen-type reactivity at the doped nitrogen sites.

Moreover, according to Table S4, the calculated formation energy for interfacial N doping in the presence of Ru nanoparticle is 0.87 eV, which is much lower than that for proximal N doping (1.91 eV). This suggests that if N doping is carried out after Ru loading, most N atoms will preferentially occupy interfacial sites. In contrast, our pre-doping strategy allows more proximal N sites to form before Ru deposition, which helps shift the reaction mechanism toward a higher contribution from the associative pathway at the early stage of the reaction. At the same time, this pre-doping strategy circumvents the Ru-induced thermodynamic bias towards interfacial N dopants during synthesis and provides a practical guideline for rationally engineering similar metal-support architectures in future studies.

The most favourable reaction pathway was selected for further validation by DFT, as shown in Figure S24. A comparison of the potential energy diagrams shows excellent agreement between MACE and DFT results, with minor discrepancies observed only for the NNH$_2$$^*$ → NNH$^*$ and NH$_2$$^*$ → NH$^*$ steps. These slight deviations suggest that the selected foundation model may exhibit limited accuracy in capturing N−H bond cleavage in this system. Nonetheless, the thermodynamic consistency is notable: the thermodynamic barrier predicted by DFT is 1.48 eV, differing by only 2.07% from that computed by MACE (1.45 eV),

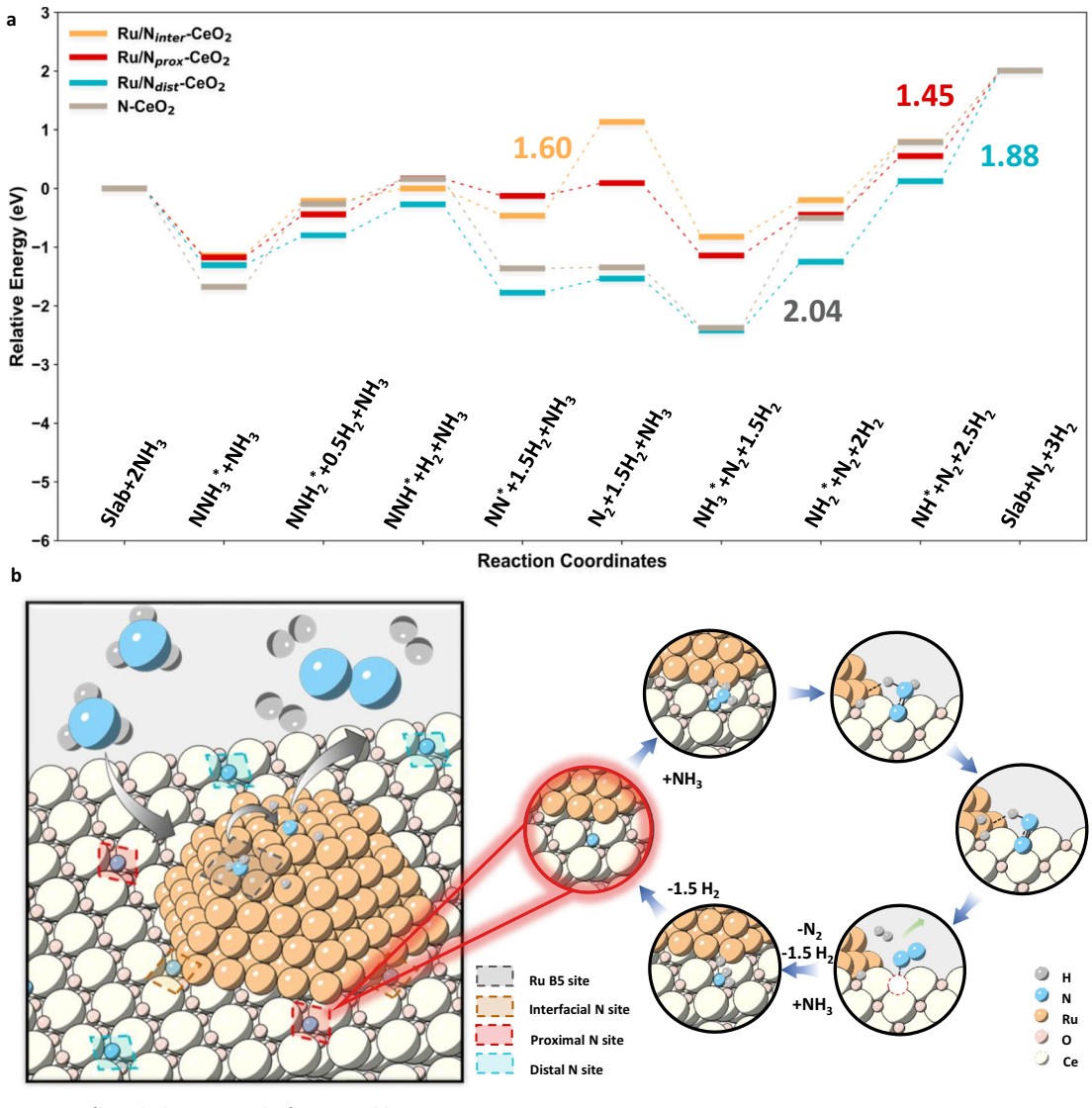

**Fig. 4 | Insights into the reaction mechanism. a** Potential energy diagrams of the MvK-type reaction at nitrogen sites under different configurations calculated by MACE. The numbers in the plot correspond to the reaction energies for the rate-determining steps in different systems. **b** Schematic representation of the concerted reaction mechanisms taking place on the dual active sites on Ru/N-CeO$_2$ catalyst. The B$_5$ site on Ru, on which the dissociative pathway proceeds, is high-lighted in grey. The N dopant active site, on which the associative pathway pro-ceeds, is highlighted in red.

highlighting the general reliability of the MACE-MP-0 foundation model.

## Discussion

The development of highly active and stable catalysts for NH$_3$ decomposition under mild condition is of great importance for hydrogen production and energy applications. Traditional dissociative NH$_3$ decomposition over metal sites suffers from the BEP constraint which establishes an inverse corelation between the atomic N binding energy and the consecutive N-H dissociation barriers. This intrinsic trade-off limits the catalytic efficiency, particularly at lower temperatures. In this study, we demonstrate that nitrogen dopants incorporated on the cerium oxide support can mimic the oxygen redox property and serve as independent active sites facilitating the associative mechanism (Fig. 4b). The nitrogen dopants were found to promote Ru site activity through electron donation and serve as anchoring sites, leading to stronger MSI, improved metal dispersion, and smaller nanoparticle sizes. Beyond these structural effects, the nitrogen dopants were also found to function as independent catalytic

centres. Using isotopic labelling experiments, it was revealed that these redox-active N sites interact with NH$_3$ reactants directly, forming NNH$_x^*$ species. Through this pathway, the energy barrier for N-H bond cleavage is thermodynamically compensated by the simultaneous formation of N−N bonds, thereby enabling enhanced catalytic activity at milder conditions.

Computational analysis further conveys the low thermodynamic barrier for the associative mechanism discussed above. Moreover, by considering three different nitrogen dopant sites, a correlation between spatial nitrogen configuration and reactivity is found. Thermo-catalytic ammonia decomposition over metal-loaded catalysts has been widely discussed as occurring at the metal-support interface due to the strong synergic MSI[49,50]. However, our results suggest that proximal nitrogen sites, located near but not directly bonded to Ru, exhibit the highest catalytic performance. These sites experience sufficient interaction with the Ru nanoparticle while avoiding overly strong adsorption of products. Mechanistically, the spatial separation between Ru and the proximal N site is sufficient to circumvent strong Ru-N binding, while still enabling the charge

transfer and polarisation needed for N-H bond breaking. Since N dopants are thermodynamically less costly at the Ru vicinity (Table S4), our pre-doping strategy, where nitrogen is introduced into the oxide lattice before metal loading facilitates the dopant configuration towards forming more proximal sites upon Ru deposition. This effectively shifts the catalyst landscape toward more favourable active site environments.

The catalysts prepared in this study inevitably contain a mixture of different N dopant configurations, making it experimentally challenging to quantify the population of each individual N site. At present, much of our understanding of proximal N sites is derived from computational analysis. Nevertheless, future work in our group is directed toward precisely engineering N dopants at controlled distances from the Ru nanoparticles. Our strategy involves forming Ru complexes coordinated with N-containing ligands positioned at well-defined spatial locations, followed by chemical deposition onto $CeO_2$ to produce catalysts with tuneable Ru–N proximities. These efforts are already in progress and are expected to provide a more definitive correlation between N-dopant positioning and catalytic behaviour. While previous studies have predominantly focused on the metal nanoparticle as the active site, our results demonstrate that N dopants on redox-active supports can also serve as genuinely potent catalytic centres, with activity levels commensurate with those of the supported Ru nanoparticle. The extra proximal N active sites in our Ru/N–$CeO_2$ catalyst circumvent the BEP scaling constraints reported for the traditional dissociative mechanism, thereby enabling $NH_3$ decomposition rates at low temperatures that rival those achieved at the metal nanoparticle. Overall, the pre-doping strategy employed here demonstrates how support engineering can be fine-tuned to enhance catalytic activity via modulating the reaction pathways. These insights provide a strong foundation for future catalyst design targeting improved hydrogen production from $NH_3$ under mild conditions.

## Methods

### Synthesis of $CeO_2$/N-$CeO_2$ support
Plain cerium oxide was synthesised using a soft-urea-glass method. Typically, 3 g of cerium chloride ($CeCl_3 \cdot 7H_2O$, 99%, Fluorochem Ltd) was mixed with 3 g urea ($CH_4N_2O$, 99.8%+, Alfa Aesar) and stirred in 60 ml of ethanol for 2 h to form a colourless homogenous gel-like solution. The mixture was then dried at 80 °C overnight to obtain a pink-red solid, followed by calcination at 800 °C for 2 h with a heating rate of 5 °C min$^{-1}$ in a tubular furnace under a 30 ml min$^{-1}$ air flow to produce the $CeO_2$ support. The nitrogen-doped $CeO_2$ supports were obtained by annealing the plain $CeO_2$ in 30 ml min$^{-1}$ $NH_3$ flow for different temperatures (T = 500, 650, 800, 900 °C) and different duration (t = 2, 4, 6, 8, 10 h) at a constant heating rate of 5 °C min$^{-1}$. The final obtained nitrogen doped cerium oxide is denoted as N-$CeO_2$ (T-t) where T stands for the calcination temperature in °C and t for the calcination time in h.

### Metal impregnation
Unless otherwise stated, the Ce-based supports were wetness impregnated with 0.01 M of tri-ruthenium dodeca-carbonyl in tetrahydrofuran solution ($Ru_3(CO)_{12}$, 99% purity (metal basis), Fisher Scientific UK Ltd). The mixture was stirred for 20 min at room temperature before drying at 80 °C overnight, followed by reduction at 450 °C for 6 h at a heating rate of 5 °C min$^{-1}$ under 5% hydrogen/nitrogen atmosphere.

### Ammonia decomposition catalytic tests
The reactor set-up consists of a gas inflow chamber in which the flow rates of gases were controlled by a Bronkhorst mass flow controller, a stainless steel fixed-bed flow reactor, and an Agilent 7890 A gas chromatograph (GC). During a typical catalysis experiment, 50 mg of catalyst was packed in a quartz tube (4 mm i.d.) and placed in the reactor. The reactor temperature was raised to a target value (ranging from 300 to 500 °C) under ambient pressure. Ammonia was introduced at controlled flow rates to achieve a range of weight-hourly-space-velocity (WHSV) from 15,000 to 72,000 ml $g_{cat}^{-1}$ h$^{-1}$ to evaluate the catalyst's performance under varying conditions. The resultant gas products were separated with an 80/100 Porapak Q column and analysed by a thermal conductivity detector (TCD) with He as the carrier gas.

### $^{15}NH_3$/$^{14}NH_3$ pulse experiment on Mass spectroscopy
The reactor setup is similar to the previously described system, consisting of an inflow gas system, a stainless-steel fixed-bed flow reactor, and an inline mass spectrometer (Hiden HPR-20 QIC Gas Analysis System). The stainless-steel gas chamber located in the inflow gas system is preloaded with 0.6 bar of either $^{15}NH_3$ ( > 98 atom%, BOC Ltd) or $^{14}NH_3$, with pressure monitored by a gauge. The preloaded gas is pulsed into the reactor while the entire system is continuously being flushed with argon at 15 ml min$^{-1}$. The final reaction products are analysed by the mass spectrometer, with the electron beam energy set to 20 eV for optimal detection. The signals of m/z = 2, 28, 29, 30, 40 are recorded. The next pulse is always pumped to the reactor once the signal of Ar (m/z = 40) reaches back to a plateau.

### Structure analysis
The lattice structure of the synthesised sample was characterised by powder X-ray diffraction (XRD) using a Brucker D8 advanced Eco X-ray diffractometer operated at 40 kV and 25 mA with Cu Kα1 radiation. The Brunauer-Emmett-Teller (BET) surface area was obtained from $N_2$ adsorption/desorption isotherms at –196 °C and analysed by a Micromeritics TriStar II Plus instrument. Active catalysts (ca. 100 mg) were pre-treated under vacuum of 10$^{-5}$ Torr at 200 °C for 18 h before taking the measurements. Actual metal loading was calculated via inductively coupled plasma mass spectrometry (ICP-MS). Nitrogen content of the support was analysed by Carbon Hydrogen Nitrogen Sulphur (CHNS) elemental analysis. Structural information on the sample surface were gained by X-ray photoelectron spectroscopy (XPS) which was performed at the EPSRC National Facility for XPS ("HarwellXPS"). The spectra were calibrated against C 1$s$ and normalised using CasaXPS software. The X-ray absorption spectroscopy (XAS) was carried out over the $K$-edge of Ru at Shanghai Synchrotron Radiation Facility (Beamline 11B); SEM were analysed using a Zeiss Sigma 300 FEG-SEM at 2 kV and 6.0 mm; STEM images and X-ray (EDX) analyses were acquired on a JEM-ARM300F, at 300 kV. The collection range of the STEM images were around 54–220 mrad.

### Temperature programmed analysis
Hydrogen temperature-programmed reduction ($H_2$-TPR) experiments were performed in an automated flow chemisorption analyser (ChemBET Pulsar). 125 mg of non-reduced catalysts were heated up to 800 °C using a temperature ramp of 5 °C min$^{-1}$ under continuous flow of 5% $H_2$/$N_2$ flow (20 ml min$^{-1}$). The $H_2$ consumption rate was monitored using a thermal conductivity detector (TCD) to interpret metal-support interaction.

Ammonia temperature-programmed desorption ($NH_3$-TPD) experiments were carried out over the same set-up as mentioned above. The catalysts, 125 mg, were treated under 20 ml min$^{-1}$ 10% $NH_3$/He flow for 2 h. The system is then heated up to 600 °C at a ramping rate of 5 °C min$^{-1}$ under continuous flush of 20 ml min$^{-1}$ He. The intensity of the ammonia peak is measured from the product stream to obtain the desorption temperature.

### *Operando* Infra-red Spectroscopy
*Operando* FTIR experiments were carried out on a Nicolet iS50 FT-IR spectrometer equipped with an MCT/A detector. In general, around 200 mg of active sample catalyst was placed in a reactor cell. After

treating the catalyst in 5% $H_2/N_2$ at 300 °C for 2 h, the background spectra were recorded at 30 °C and 10 ml min$^{-1}$ Ar flow. The reactor cell was heated up from room temperature to 400 °C with an increment of 50 °C under continuous 10% $NH_3$/He flow (10 ml min$^{-1}$). Each spectrum was averaged with 64 scans at a resolution of 4 cm$^{-1}$.

IsP-FTIR experiments were carried out on the same setup at 400 °C. The background was taken at 400 °C and 10 ml min$^{-1}$ Ar flush. Each pulse contained 0.6 bar of either $^{15}NH_3$, $^{14}ND_3$ or $^{14}NH_3$. The spectrum was continuously taken for 20 min after each pulse to ensure all intermediates and surface species are flushed away and the spectrum restored to the base line. Spectra were recorded with a time interval of 30 s and averaged by 32 scans.

## Computational Details

In this study, the reaction mechanism has been simulated using the Machine Learning-based Atomic Cluster Expansion (MACE) framework, and further verified using DFT. The publicly available MACE-MP-0 foundation model−trained on the Materials Project dataset was employed for its demonstrated reliability in describing transition-metal oxides[51–53]. Reaction pathways for $NH_3$ decomposition at various surface sites were first explored at the MACE level, and the most thermodynamically favoured MvK pathway was subsequently validated using DFT to ensure the accuracy of MACE. All MACE calculations were conducted with a force convergence tolerance of 0.02 eV Å$^{-1}$, matching the DFT set-up. DFT calculations were carried out using the Vienna Ab initio Simulation Package (VASP)[54] 5.4.4. The exchange−correlation functional was treated within the generalised gradient approximation (GGA) using the Perdew−Burke−Ernzerhof (PBE) form[55]. Van der Waals interactions were incorporated via the DFT-D3 method by Grimme[56]. A plane-wave cutoff energy of 400 eV and a $3 \times 2 \times 1$ Monkhorst-Pack $k$-point mesh were used for all slab calculations. Geometry optimisations were considered converged when the total energy change was below 10$^{-4}$ eV and the Hellmann−Feynman forces were lower than 0.02 eV Å$^{-1}$. To correctly account for the localised 4$f$ electrons of cerium, a Hubbard U correction of 4.5 eV was applied using the 'Dudarev' approach, consistent with prior studies[57–59]. The climbing-image nudged elastic band (CI-NEB) method was used to search for the minimum energy pathway for proximal doped N migration to Ru cluster (Figure S23), with the same setting of convergence to keep data consistency.

The cluster-loaded slab models used in the simulations were developed with reference to both experimental data and prior literature reports. The exposed $CeO_2$ (110) surface has been observed in experimental TEM, in agreement with previous studies reporting that the (110) surface exhibits a strong propensity for oxygen vacancy formation and thus nitrogen doping[58,59]. Accordingly, a $3 \times 3$ $CeO_2$ (110) surface slab, consisting of six atomic layers and a vacuum of 25 Å thickness along the z-direction, was constructed from the optimised bulk structure. A $Ru_{10}$ cluster was chosen as a representative model of the actual Ru NPs in accordance with both experimental results from TEM and XAS, as well as prior literature demonstrating the structural stability and prevalence of $Ru_{10}$ clusters as catalysts in surface science models[53,60]. Notably, the binding energy of the $Ru_{10}$ cluster on the $CeO_2$ (110) surface is calculated to be 5.82 eV in our model, indicating its strong anchoring and the inferred stability of the model cluster[61–65]. All N-doped slab models were constructed by substituting a single surface oxygen atom with a nitrogen atom, in consideration of the low doping concentration (Figures S25 and S26).

## Data availability

All data supporting the findings of this study are available within the paper and the Supplementary Information. Data that support the findings of this manuscript are also available from the corresponding author upon request.

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

## Acknowledgements
D.Y. and S.C.E.T. acknowledge financial support from the EPSRC Divisional CASE Conversion Incentivisation Scheme and OXGRIN, plc. The X-ray photoelectron (XPS) data collection was performed at the EPSRC National Facility for XPS ("HarwellXPS"), operated by Cardiff University and UCL, under Contract No. PR16195. M.S. and M.L. gratefully acknowledge the John Fell Oxford University Press Research Fund (application #0015331), which sponsored the hardware on which the computational part of the work was performed.

## Author contributions
D.Y. performed the experiments and the manuscript preparation. M.L. conducted the computational part of the study. X.L. and W.L. collected and analysed (S)TEM and EDS results. M.D. performed the SEM measurements. C.F. carried out the high resolution XPS. X.P. and J.L. assisted with the isotopic experiments. Y.L., S.W. and M.S. revised the manuscript. All the authors discussed and contributed further edits to the paper. M.S. supervised the computational part of the work. S.C.E.T. conceived and designed the research.

## Competing interests
The authors declare no competing interests.
