## [Transparent Peer Review file · Nature Communications]

Unravelling the Role of Redox Active Sites in Nitrogen Doped Cerium Oxide for Associative Ammonia Decomposition

Corresponding Author: Professor Michail Stamatakis

Version 0:

Reviewer comments:

Reviewer #1

(Remarks to the Author)

Reviewer #2

(Remarks to the Author)

In this paper, CeO₂ was synthesised using a typical soft urea glass (SUG) method, while its nitrogen-doped derivatives were obtained by subsequent annealing under NH₃ flow at varying temperatures and durations. Ruthenium was loaded via wetness impregnation. This technical approach is inspired by biomimetic concepts from nature, demonstrating strong innovativeness and practical applicability.

The authors employed a series of characterization techniques and activity tests to demonstrate that calcination at 650°C under NH₃ atmosphere facilitates the formation of a superior support material, which is also reflected in its efficiency. Under this synthesis condition, the ammonia decomposition efficiency remains high, with the conversion energy barrier reaching an ideal level. Through in-situ DRIFTS of ammonia reaction and DFT calculations, the authors confirmed that this synthesis method effectively relax or overcome BEP constraints.

In Figure S16, the authors attribute the peak at 3334 cm⁻¹ to the decomposition reaction. However, the temperature range investigated is only 25-400°C. Based on the reaction efficiency reported by the authors, this stage should correspond to NH₃ desorption, which is also consistent with the NH₃-TPD results. If the authors insist that the attenuation of the peak in this region is caused by the decomposition reaction, reasonable evidence should be provided. Additionally, since the efficiency tests in this paper are conducted above 400°C, I believe in-situ DRIFTS data in this higher temperature range should be supplemented, as the reaction performance at 400°C is not particularly satisfactory.

The authors conducted a detailed and systematic investigation of the support, proposing that loading Ru on N-CeO₂ can effectively lower the energy barrier for N-H dissociation, which is the rate-determining step. DFT calculations were employed to analyze reaction energy barriers and related data. However, have the authors studied catalysts with different Ru loadings? In this reaction, is it possible to determine the relative contributions of the support and the active metal? Specifically, to what extent can nitrogen doping minimize the required Ru loading?

Aside from these two questions, I find that the paper presents substantial experimental data and draws inspiration from biomimetic concepts, demonstrating considerable innovation.

Reviewer #3

(Remarks to the Author)

The manuscript describes a catalyst for ammonia decomposition with enhanced activity due to the use of a Nitrogen-doped ceria in comparison to the non-doped support. Besides evaluating the catalytic performance, the mechanism is studied by operando NH₃ DRIFTS, isotopic exchange, machine-learning and DFT.

Apart from that, there are some issues regarding the experimental results and characterization and some of the conclusions regarding the mechanism proposed are hypothetical and not completely supported by the experimental characterization and

literature. As it stands, I cannot recommend it for publication in Nature communications. In the following are my comments/questions on some specific issues that should be addressed by the authors:

In line 192, the average particle size is 2.2 nm. However, in supplementary information Figure S14 and EDS mapping, most of particles are larger, around 4-5 nm.

In line 155 it says: "the main contribution of catalytic activity arises from Ru". This seems to be with strong contradiction with the larger part of the manuscript, which is devoted to the study of the associative mechanism induced by proximal N sites. Even the associative mechanism is mentioned in the title. To be more informative, it would help the quantification of the relative contribution to the activity of to each type of site, either Ru (dissociative mechanism) or N sites (associative mechanism).

Some effects of N-doping of carbon materials are well known and widely reported in the literature but not sufficiently supported by a literature revision in this manuscript. For instance, these well-known effects are:

- "help to stabilise the Ru and inhibit further aggregation" (line 187)

- it is described a peak shift observed in the spectrum (in line 182)

- "Nitrogen doping also facilitates electron transfer..." (in line 133)

There are two references (31, 32) about N-doped carbon material characterization but they are not about ammonia decomposition. Relevant references about N-doping metal interaction (ACS Catal. 5 (2015) 2740–2753.

<https://doi.org/10.1021/acscatal.5b00094>) and NH₃ decomposition are lacking (J. Phys. Chem. C 116 (2012) 26385–26395).

The N-doping on a conductive support such as carbon can have different effects of N doping of a metal oxide which have lower electron conductivity. Therefore, references about N-doping of Ce would be also relevant.

What is the actual Ru content for N-CeO₂ and CeO₂?

In line 163, the increase of reduction temperature along the N content is explained because N facilitates electron transfer. If facilitates electron transfer, it makes sense that the reduction should be favoured. In fact, Ru peak shift to more reduced species in the XPS spectrum.

Minor comments

In line 48 it says "inverse correlation between the NH₃ binding energy..." It should be N₂ instead of NH₃ to be consistent also with the abstract line 23. Some reference could be added to support this.

It is not clear what N- proximal is. Is it equivalent to lattice N?. The N proximal site should be defined or depicted in a Figure like 4.

Consider moving Table S2 of the comparison into the main article. Moreover, since Ru is expensive, comparison with other less expensive transition metals would be also important.

Version 1:

Reviewer comments:

Reviewer #1

(Remarks to the Author)

The authors have revised the manuscript according to my comments and it can be accepted as it is.

Reviewer #2

(Remarks to the Author)

The author of this paper has provided a rigorous and scientific response to the reviewer's comments. The author has provided reasonable answers to the comments I raised, and I believe the current status of the paper is acceptable.

Reviewer #3

(Remarks to the Author)

The authors have addressed all my comments satisfactorily

Point-to-point Response to Reviewers' Comments

Manuscript ID: NCOMMS-25-74873-T

Title: Unravelling the Role of Redox Active Sites in Nitrogen Doped Cerium Oxide for Associative Ammonia Decomposition

General response: We sincerely thank the editor and all Reviewers for their thorough evaluation of our manuscript and for the constructive and positive comments provided. We have carefully considered each point and revised the manuscript accordingly. These revisions have substantially improved the clarity and overall quality of the work. A detailed, point-to-point response is provided below. In this response letter, Reviewers' comments are shown in black, and our responses are presented in blue. All the changes in our revised manuscript and Supplementary Information are highlighted in yellow.

Reviewers' comments

Reviewer #1:

This manuscript presents an exceptional study on the role of nitrogen dopants as independent active sites for ammonia decomposition. The work is original, demonstrating convincingly that N-dopants can facilitate an associative Mars-van Krevelen (MvK) mechanism, thereby circumventing the limitations imposed by classical Brønsted-Evans-Polanyi (BEP) scaling relationships. The combination of experimental techniques, such as the isotopic labelling studies, with advanced theoretical calculations (MACE and DFT) provides evidence. This work is worthy publication if the authors can address the following points to further enhance its clarity and impact.

General response: We sincerely appreciate this Reviewer's careful evaluation of our manuscript and his/her positive assessment of the originality and significance of our work. The constructive comments have helped us improve the clarity, depth, and impact of the manuscript. Accordingly, we have revised the manuscript by expanding the mechanistic discussion, clarifying the definition and role of proximal N sites, refining the presentation of figures, and providing additional explanation for our computational model. We believe these revisions have strengthened both the experimental and theoretical aspects and have addressed all concerns raised. Detailed responses to each comment are provided below.

(1). The manuscript excellently describes the creation of N-vacancies (V_N) during the MvK cycle but leaves the regeneration mechanism somewhat implicit. The description of N-sites as

"redox-active" is central to the analogy with enzymes and ceria's oxygen redox chemistry, but the specific redox couple (e.g., $N^{3-} \leftrightarrow N_2$) and the re-filling of V_N could be elaborated.

Response: We sincerely thank the reviewer for this insightful comment. In the original manuscript, we described the formation of nitrogen vacancies (V_N) during the MvK cycle, but the regeneration process and associated nitrogen redox couple were not elaborated upon with sufficient clarity. Therefore, in the revised manuscript, we now explicitly describe the complete Mars-van Krevelen cycle for NH_3 decomposition on Ru/N-CeO₂ in the *Theoretical insights* section. This revision clarifies how lattice nitrogen (N^{3-}) participates directly in N_2 formation, how V_N is generated, and how it is subsequently replenished by nitrogen from NH_3 , thereby making the full N^{3-}/N^0 redox cycle explicit.

Revisions:

This figure is not changed but included here for the reviewer's convenience:

Fig. 4a. Potential energy diagrams of the MvK-type reaction at nitrogen sites under different configurations calculated by MACE. The numbers in the plot correspond to the reaction energies for the rate-determining steps in different systems.

The following text was added to the "Theoretical insights ..." section on page 10:

“In this MvK cycle, the doped lattice nitrogen acts as a redox-active reservoir. The cycle starts from a fully occupied N_{doped} site. Successive dehydrogenation of NH_3 produces surface NH_x intermediates, which couple with the lattice N_{doped} to eventually form N_2 . These steps progressively oxidise the lattice nitrogen N^{3-} to N^0 and generate a nitrogen vacancy (V_N). The N_2 molecule then desorbs from the Ru/N-CeO₂ surface with an associated thermodynamic cost (NN^* to N_2). Subsequently, nitrogen from incoming NH_3 refills V_N via further dehydrogenation and N insertion, restoring the N_{doped} site (NH_3^* to slab). Together, N_2 desorption and V_N refilling complete the MvK cycle and establish the lattice N sites as genuinely redox-active.”

(2). The "proximal N-site" is a key concept for the highest activity, but its definition ("located near but not directly bonded to Ru") remains qualitative. Readers may wonder about the experimental evidence for its population and how it can be engineered.

Response: We thank the reviewer for this important point. We fully agree that a quantitative, direct experimental mapping of individual N-dopant positions relative to Ru nanoparticles would be highly desirable, but is extremely challenging with current characterisation techniques. In the revised manuscript, we therefore clarify more explicitly how proximal N sites are defined, how their catalytic role is established, and how their formation is rationalised within our synthesis strategy.

Although the direct imaging of Ru-N distances is not experimentally feasible at present, we have employed a combination of computational analysis and experimental observations that consistently confirm the substantial population of proximal N sites and identify them as the most active. Computationally, we calculated both the formation energies of different N-dopant configurations and the rate-determining step (RDS) energy barriers (ΔE_{RDS}) as a function of the Ru-N distance. These calculations show that N dopants are thermodynamically favoured in the vicinity of Ru (Table S4). In response to the reviewer's comment on a clearer definition, we have added Fig. S22, which illustrates the ΔE_{RDS} as a function of the Ru-N distance. The results reveal a non-monotonic dependence: the ΔE_{RDS} decreases from the interfacial site (Ru-N distance ≈ 1.7 Å), reaches a minimum at the proximal site (≈ 4.1 Å), and then increases again for distal or Ru-free configurations. This trend identifies the proximal site as the optimal configuration and suggests a clear sense of the relevant spatial range.

In addition, we have clarified in the revised text that our **pre-doping strategy** biases dopants toward proximal rather than interfacial locations. Introducing N into the oxide lattice prior to Ru deposition avoids the strong thermodynamic driving force for forming primarily interfacial N (Table S4), which would indeed be preferred if N doping was performed after metal loading. At the same time, we explicitly acknowledge that the synthesised catalyst inevitably contains a distribution of various N-dopant environments, and isolating only one configuration is not currently achievable. This limitation and its implications for rational synthesis design are now stated more clearly in the revised manuscript.

Moreover, it should be noted that several experimental observations, including H₂-TPR, XAS, IsP-FTIR, and pulsed MS experiments collectively support the computational observation of proximal N sites. These datasets, while not capable of directly resolving individual atomic distances, are most consistently interpreted by a model in which a population of N atoms located near, but not too close to, Ru nanoparticles participate in the associative MvK pathway and thus enhances catalytic activity.

Furthermore, in response to the Reviewer's comment, we explicitly outline feasible routes for future validation and control of Ru-N spatial arrangements. These include synthesising Ru complexes with N-containing ligands positioned at well-defined distances prior to deposition, as well as advanced structural probes such as high-resolution TEM, atom-probe tomography, and synchrotron/neutron-based methods capable of distinguishing local N environments. These

directions have been added to the revised manuscript as a roadmap for further establishing the structure-activity relationship.

We hope these clarifications improve the manuscript and address the Reviewer’s concerns by explaining how proximal N sites are expected to form under our synthesis conditions and why both computation and experiment consistently identify them as the most catalytically relevant configuration.

Revisions:

The following text was added to the “Theoretical insights...” section, page 11:

“At the same time, this pre-doping strategy circumvents the Ru-induced thermodynamic bias towards interfacial N dopants during synthesis and provides a practical guideline for rationally engineering similar metal-support architectures in future studies.”

The following text was added to the “Discussion” section, page 13:

“The catalysts prepared in this study inevitably contain a mixture of different N dopant configurations, making it experimentally challenging to quantify the population of each individual N site. At present, much of our understanding of proximal N sites is derived from computational analysis. Nevertheless, future work in our group is directed toward precisely engineering N dopants at controlled distances from the Ru nanoparticles. Our strategy involves forming Ru complexes coordinated with N-containing ligands positioned at well-defined spatial locations, followed by chemical deposition onto CeO₂ to produce catalysts with tuneable Ru–N proximities. These efforts are already in progress and are expected to provide a more definitive correlation between N-dopant positioning and catalytic behaviour.”

The following figure was added to the SI, page 23:

Fig. S22 Ru–N_{doped} distance–dependent modulation of the rate-determining step barrier of MvK type NH₃ decomposition on Ru/N-CeO₂.

The following text was added to the “Theoretical insights...” section, pages 10-11:

“As shown in Fig. S22, the calculated RDS barrier exhibits a non-monotonic dependence on the Ru-N_{doped} distance: it drops from the interfacial (Ru-N bond distance 1.7 Å) to the proximal N site (Ru-N distance 4.1 Å) but increases again for the distal and Ru-free CeO₂ cases. This non-monotonic trend identifies the proximal N configuration as the optimum, giving the lowest RDS barrier and thus the highest predicted activity.”

(3). The coexistence of the dissociative (on Ru) and associative (on N) pathways is well-established. However, a natural question is their relative contribution to the overall superior activity.

Response: We thank the reviewer for highlighting this important question regarding the relative contributions of the associative and dissociative pathways. In response to this comment, we used integrated peak areas from our isotopic pulse MS measurements to monitor the formation of ¹⁵N-¹⁵N (m/z = 30) and ¹⁵N-¹⁴N (m/z = 29). Continuous ¹⁵NH₃ feeding, which would enable steady-state quantification, was not feasible due to the high cost and limited availability of ¹⁵NH₃. The pulsed-MS approach, however, provides reliable mechanistic insight because each isotopic pulse captures transient N₂ formation processes. Upon a ¹⁵NH₃ pulse, the m/z = 30 signal reflects N₂ generated solely through the dissociative pathway on Ru sites, whereas the m/z = 29 signal arises from coupling between ¹⁵N from the feed and ¹⁴N from the ceria lattice, thus serving as a direct fingerprint of the associative MvK-type pathway. Importantly, repeated pulsing behaviour further confirms this assignment: after a ¹⁵NH₃ pulse, the subsequent ¹⁴NH₃ pulse also yields m/z = 29, indicating that lattice nitrogen undergoes isotopic exchange (via formation and refilling of N vacancies), consistent with the proposed associative mechanism. After careful baseline correction and integration of the transient isotopic signals, the isotopic pulse MS analysis shows that, for the representative Ru/N-CeO₂ (650-6) catalyst at 450 °C, approximately 36.7 ± 1.5% of N₂ formation proceeds via the associative pathway, while 63.3 ± 1.5% arises from the dissociative pathway. These results demonstrate that both mechanisms operate concurrently under reaction conditions, with the associative pathway becoming particularly significant at lower Ru loadings where more N-doped sites remain accessible.

The new analyses have been incorporated into the revised manuscript, and the updated discussion is provided below for the Reviewer’s convenience.

Revisions:

The following sentence was added to the “In-situ analysis and mechanistic understanding” section on page 10 of the main manuscript:

“By quantifying the area of the peaks corresponding to ¹⁴N-¹⁵N and ¹⁵N-¹⁵N during each ¹⁵NH₃ pulse, it is revealed that during the catalytic reaction, approximately 36.7 ± 1.5% of the products arises from the associative pathway, while 63.3 ± 1.5% originate from dissociative mechanisms.”

(4). Figure 3g: Clearly annotating on the figure which pulses are $^{14}\text{NH}_3$ and which are $^{15}\text{NH}_3$ would improve clarity.

Response: We thank the reviewer for this helpful suggestion. In the revised Figure 3g, we have clearly annotated the pulses corresponding to $^{14}\text{NH}_3$ and $^{15}\text{NH}_3$ to improve clarity and facilitate interpretation of the isotopic signals. We have also added labels indicating the relative contributions of the associative and dissociative pathways, based on the integrated peak intensities of $^{15}\text{N}-^{14}\text{N}$ ($m/z = 29$; associative) and $^{15}\text{N}-^{15}\text{N}$ ($m/z = 30$; dissociative).

Revision:

The updated Fig. 3g can be found on page 9 of the revised manuscript and is also copied below:

Fig. 3g. $^{15}\text{NH}_3/^{14}\text{NH}_3$ pulse experiment on MS. Operated at 450 °C, on Ru/N-CeO₂ (650-6) under 10 ml min⁻¹ flow of Ar, each pulse contains 0.6 bar of either $^{15}\text{NH}_3$ or $^{14}\text{NH}_3$.

(5). Figure 4b: The schematic is helpful but could be enhanced. Consider using different colored arrows or labels to distinctly differentiate the dissociative pathway on the Ru B5 site from the associative MvK pathway on the N-site. This would immediately visually reinforce the dual-pathway concept.

Response: We thank the reviewer for the valuable suggestion. In the revised Fig. 4b, we now use colour-coded arrows to clearly distinguish the dissociative pathway on Ru sites (grey) from

the associative MvK pathway on N sites (red). We have also used different colours for the various N sites to better illustrate their spatial distinctions. We believe these changes have improved the visual clarity of the dual-pathway concept.

Revision:

The edited Fig. 4b (page 12 in the revised manuscript) is shown below:

Fig. 4b. Schematic representation of the concerted reaction mechanisms taking place on the dual active sites on Ru/N-CeO₂ catalyst. The B₅ site on Ru, on which the dissociative pathway proceeds, is highlighted in grey. The N dopant active site, on which the associative pathway proceeds, is highlighted in red.

(6). In the computational model section, it is highly suggested to conduct the MD simulations to make sure whether the constructed cluster are structurally stable and reasonable under the reaction conditions. This will directly impact the subsequent mechanistic study and the conclusion.

Response: We sincerely appreciate the reviewer’s thoughtful suggestion. Molecular dynamics (MD) simulations could, in principle, provide valuable insight into possible structural reconstruction of supported Ru nanoparticles under reaction conditions, and such information would indeed strengthen the assessment of cluster stability. However, carrying out statistically meaningful, DFT-level MD simulations capable of reliably sampling Ru-cluster reconstruction along the NH₃ decomposition pathway would require exceedingly long simulation times and extensive configurational sampling, which are beyond the scope of this study and the computational resource capacity available for this study. Still though, previous computational studies have established the stability of supported Ru_n clusters, and our representative calculations and experimental data provide further evidence for the stability thereof.

In particular, in line with established practice in prior theoretical investigations of Ru_n/CeO₂ catalysts, including studies on the nitrogen reduction reaction, NH₃ decomposition, dry reforming of methane, and CO₂ methanation (citations included in the ‘Revisions’ part below), we adopt a fully relaxed Ru₁₀/CeO₂ model as a representative small-cluster system. Supported clusters of comparable size (including Ru₁₀) have been widely used in the literature without the need for explicit finite-temperature MD, and we now make this modelling rationale clearer in the revised manuscript. To further support the structural robustness of our chosen model, we report the calculated binding energy of the Ru₁₀ cluster on the CeO₂(110) surface (-5.82 eV), which indicates strong anchoring and provides confidence to the stability of the interface under reaction conditions.

Importantly, our computational model is grounded in experimental observations: TEM and EXAFS analyses confirm the Ru particle size and dispersion (main text Fig. 2 and Fig. S15), and the catalyst exhibits excellent stability over 70 hours on stream (main text Fig. 1e), consistent with a structurally robust Ru/CeO₂ system. While we acknowledge that the real catalyst contains a distribution of Ru cluster sizes and morphologies, the mechanistic aim of this work is to disentangle and quantify how the Ru-N spatial distance influences the reaction energetics. To keep this variable under control, we therefore focus on a single, representative Ru₁₀ cluster and systematically vary only the position of the N dopant.

To address the Reviewer’s comment more clearly, we have (i) added citations to prior DFT studies employing supported Ru_n clusters as realistic catalytic models, and (ii) incorporated the Ru₁₀ binding energy value (-5.82 eV) to document the strong anchoring and inferred stability of the model cluster. Together with the experimental stability data, we believe these additions provide sufficient justification for the modelling approach adopted here, while we also recognise that a full finite-temperature MD investigation of cluster reconstruction remains an important direction for future work.

Revisions:

The following sentence has been added in section “Computational Details” on page 15 of the manuscript:

“Notably, the binding energy of the Ru₁₀ cluster on the CeO₂ (110) surface is calculated to be -5.82 eV in our model, indicating its strong anchoring and the inferred stability of the model cluster.⁶⁹⁻⁷³”

The following references have been added to support this clarification:

69. Cao, H. & Zhou, S. Atomic Ru clusters supported on CeO₂(110) for effectively catalyzing the electrochemical N₂ reduction reaction: insights from density functional theory. *New J. Chem.* 48, 5919–5929 (2024).
70. Le, T. A. et al. Ru dispersed on CeO₂{1 0 0} facets boosting the catalytic NH₃ decomposition for green H₂ generation. *Chem. Eng. J.* 493, 152503 (2024).

71. Qu, P. F. & Wang, G. C. DFT-based microkinetic model analysis of dry reforming of methane over Ru₇/CeO₂(111) and Ru₇/CeO₂(110): key role of surface lattice oxygen vacancy. *Catal. Sci. Technol.* 12, 1880–1891 (2022).
72. Piotrowski, M. J., Tereshchuk, P. & Da Silva, J. L. F. Theoretical investigation of small transition-metal clusters supported on the CeO₂(111) surface. *J. Phys. Chem. C* 118, 21438–21446 (2014).
73. Liu, G. et al. Crystal facet regulation of Ru/CeO₂ catalysts towards boosted low-temperature CO₂ methanation. *J. Environ. Chem. Eng.* 13, 115700 (2025).

Reviewer #2:

In this paper, CeO₂ was synthesised using a typical soft urea glass (SUG) method, while its nitrogen-doped derivatives were obtained by subsequent annealing under NH₃ flow at varying temperatures and durations. Ruthenium was loaded via wetness impregnation. This technical approach is inspired by biomimetic concepts from nature, demonstrating strong innovativeness and practical applicability.

The authors employed a series of characterization techniques and activity tests to demonstrate that calcination at 650°C under NH₃ atmosphere facilitates the formation of a superior support material, which is also reflected in its efficiency. Under this synthesis condition, the ammonia decomposition efficiency remains high, with the conversion energy barrier reaching an ideal level. Through in-situ FTIR of ammonia reaction and DFT calculations, the authors confirmed that this synthesis method effectively relax or overcome BEP constraints.

General response: We sincerely thank the reviewer for the positive assessment of our study and for recognising the novelty and practical relevance of the proposed catalytic strategy. We also appreciate the constructive comments, which have helped us improve the clarity and completeness of the manuscript. In response, we have revised the text to better justify the interpretation of the *operando* FTIR data, expand the discussion on the impact of Ru loading and the role of N-doping, and further clarify several experimental and mechanistic aspects. Additional data and discussions have been incorporated. We believe these revisions have significantly strengthened the manuscript.

(1). In Figure S16, the authors attribute the peak at 3334 cm⁻¹ to the decomposition reaction. However, the temperature range investigated is only 25-400°C. Based on the reaction efficiency reported by the authors, this stage should correspond to NH₃ desorption, which is also consistent with the NH₃-TPD results. If the authors insist that the attenuation of the peak in this region is caused by the decomposition reaction, reasonable evidence should be provided. Additionally, since the efficiency tests in this paper are conducted above 400°C, I believe in-situ DRIFTS data in this higher temperature range should be supplemented, as the reaction performance at 400°C is not particularly satisfactory.

Response: We thank the reviewer for this insightful and constructive comment, and we apologise for any confusion caused in the original description. To clarify, the 3334 cm⁻¹ band in Fig. S18 in the updated SI [previously S16] corresponds to the N-H stretching modes of NH₃, arising from both gaseous NH₃ and NH₃ adsorbed on the catalyst surface, with the signal predominantly contributed by gaseous NH₃. The FTIR data were collected under *operando* conditions, with the measurement cell continuously purged by 10 mL min⁻¹ of 10% NH₃/He. Therefore, if only desorption were occurring, the intensity of the 3334 cm⁻¹ band would not be expected to decrease, as the overall population of N-H species in the cell would remain unchanged and the gaseous-phase NH₃ concentration would remain fixed at 10% under the constant-flow conditions.

By contrast, when NH₃ decomposition takes place, NH₃ within the measurement cell is converted to N₂ and H₂, reducing the total number of N-H species. In this scenario, a measurable

decline in the 3334 cm^{-1} band is expected, which is consistent with the trends observed in our *operando* spectra. In another word, the attenuation of this band reflects a genuine reduction in (gaseous) NH_3 concentration in the measurement cell arising from its decomposition. This interpretation supports our conclusion that NH_3 decomposition becomes significant at elevated temperatures, in line with prior literature. To further substantiate this interpretation, we have amended Supplementary discussion 5 [previously Supplementary discussion 4], to explain, in reference to Fig. 3b that at elevated temperatures, the broad O-H band that appears following a $^{14}\text{NH}_3$ pulse is consistent with hydrogen produced from NH_3 decomposition reacting with surface oxygen to form surface hydroxyl groups. The emergence of this O-H feature provides further confirmation of NH_3 decomposition under the *operando* conditions.

Regarding the Reviewer's suggestion to provide FTIR data above 400 $^\circ\text{C}$, we sincerely apologise that this is currently not feasible due to the design and safety constraints of our *operando* FTIR setup. The current setup has an upper temperature limit of 400 $^\circ\text{C}$, which is one of the reasons our measurements were performed up to this temperature. In addition, the configuration requires approximately 200 mg of catalyst and a flow of 10 mL min^{-1} of 10% NH_3/He , corresponding to a weight hourly space velocity (WHSV) of $\sim 3,000 \text{ mL g}^{-1} \text{ h}^{-1}$, which is substantially lower than the WHSV used in actual catalytic tests (15,000 $\text{mL g}^{-1} \text{ h}^{-1}$). A lower WHSV would result in a longer residence time, which in turn would greatly increase NH_3 conversion. Therefore, performing FTIR above 400 $^\circ\text{C}$ would likely lead to very fast and near-complete NH_3 conversion, making it difficult to capture meaningful mechanistic information or resolve surface intermediates (*i.e.*, no 3334 cm^{-1} band could be observed when NH_3 fully decomposes). For this reason, obtaining *operando* FTIR data in a lower-conversion regime is more appropriate for mechanistic interpretation in this system.

We would also like to clarify that the aim of the FTIR experiments was to demonstrate the temperature-dependence of NH_3 decomposition rather than to quantify catalytic performance. We hope the reviewer agrees that the trends observed in Fig. S18 are sufficient to support this conclusion.

In view of the above observations, we have revised the discussion in the main text and added clarifications to the Supplementary Information (Supplementary Discussion 5) to ensure transparency and to help correctly interpret the *operando* spectra within the practical constraints of the technique.

Revisions:

The revised discussion is copied below, which can be found in section “Supplementary discussion 5: Operando DRIFT analysis” on page 36 of the Supporting Information:

“Four characteristic NH_3 vibrational features are observed in the FTIR spectra: two symmetric N-H bending modes at 930.5 and 966.6 cm^{-1} , an asymmetric N-H bending mode at 1627 cm^{-1} , and a symmetric N-H stretching mode at 3334 cm^{-1} (Figs. S17–S18).¹³ The FTIR measurements were conducted under *operando* conditions, with the catalyst continuously purged by 10 mL min^{-1} of 10% NH_3/He . Under these conditions, the 3334 cm^{-1} band reflects contributions from both gaseous NH_3 and NH_3 adsorbed on the catalyst surface, with the gaseous-phase component

dominating the overall intensity. The intensity of the 3334 cm^{-1} band decreases for Ru-impregnated catalysts as temperature increases (Fig. S18). This attenuation cannot be attributed to NH_3 desorption, as desorption alone would not alter the total population of N-H species in the measurement cell. Instead, the decline in intensity signifies the onset of NH_3 decomposition, in which NH_3 is converted to N_2 and H_2 , reducing the number of N-H species contributing to the signal. This temperature-dependent behaviour aligns with reports from the literature on thermocatalytic NH_3 decomposition.¹⁴ Additional evidence supporting NH_3 decomposition is provided by the appearance of a broad O-H band during the $^{14}\text{NH}_3$ -pulsed FTIR experiments at elevated temperatures (Fig. 3b). The formation of hydrogen during NH_3 decomposition facilitates the generation of surface hydroxyl species through reaction with lattice oxygen, producing O-H vibrational features that diminish upon purging, consistent with hydroxyl formation. At lower temperatures, a sharp band at 1400-1425 cm^{-1} is observed, corresponding to NH_3 adsorption on CeO_2 surface sites.^{15,16}

It should be noted that the *operando* FTIR cell used in this study has a maximum operating temperature of 400 °C, which defines the upper limit of the measurements. The high catalyst mass required (~200 mg) and relatively low flow rate (10 mL min^{-1} of 10% NH_3/He) correspond to a weight hourly space velocity (WHSV) of ~3000 $\text{mL g}^{-1} \text{h}^{-1}$, much lower than that used in catalytic performance tests (15,000 $\text{mL g}^{-1} \text{h}^{-1}$). Under such low WHSV conditions, increasing the temperature above 400 °C would likely result in near-complete NH_3 conversion, making the detection of intermediates and mechanistic features exceedingly difficult. For this reason, *operando* FTIR measurements at temperatures above 400 °C are not suitable for mechanistic analysis in this system, as the NH_3 stretching band at 3334 cm^{-1} would no longer be observable. The trends observed in Figs. S17-S18 reliably demonstrate the enhanced decomposition of NH_3 at elevated temperatures and support the mechanistic conclusions discussed in the main text.”

(2). The authors conducted a detailed and systematic investigation of the support, proposing that loading Ru on N- CeO_2 can effectively lower the energy barrier for N-H dissociation, which is the rate-determining step. DFT calculations were employed to analyze reaction energy barriers and related data. However, have the authors studied catalysts with different Ru loadings? In this reaction, is it possible to determine the relative contributions of the support and the active metal? Specifically, to what extent can nitrogen doping minimize the required Ru loading?

Response: We thank the reviewer for this insightful comment. To address this point, we synthesised additional catalysts with systematically varied Ru and N contents, and performed catalytic tests accordingly. As shown in Fig. S7, the NH_3 conversion achieved by Ru/N- CeO_2 (650-6) (Ru content: 0.3 wt.%; N content: 3.1 at.%) is ca. 38%, which is comparable to that of Ru/ CeO_2 (Ru content: 0.9 wt.%; N content: 0 at.%), ca. 35% under identical conditions (i.e., 450 °C, WHSV = 30 000 $\text{mL g}_{\text{cat}}^{-1} \text{h}^{-1}$). This demonstrates that nitrogen doping can reduce the required Ru loading by nearly a factor of three while maintaining similar catalytic performance.

Moreover, the promotional effect of N doping is most pronounced at lower Ru loadings. At these loadings, a substantial fraction of the support surface remains exposed, enabling N-doped sites to actively participate in the associative reaction pathway. As the Ru loading increases, Ru increasingly covers the support surface, reducing the number of accessible N sites, leading to

the dominance of the dissociative pathway on Ru B₅ sites. This trend is now discussed in the revised Supplementary Discussion 4.

Regarding the quantification of the relative contributions from the support (associative pathway) and the Ru sites (dissociative pathway), we used integrated peak areas from our isotopic pulse MS measurements to monitor the formation of ¹⁵N-¹⁵N (m/z = 30) and ¹⁵N-¹⁴N (m/z = 29). Continuous ¹⁵NH₃ feeding, which would allow steady-state quantification, was not feasible due to the high cost and limited availability of ¹⁵NH₃. The pulsed-MS approach, however, provides reliable mechanistic insight because each isotopic pulse captures transient N₂ formation processes. Upon a ¹⁵NH₃ pulse, the m/z = 30 signal reflects N₂ generated solely through the dissociative pathway on Ru sites, whereas the m/z = 29 signal arises from coupling between ¹⁵N from the feed and ¹⁴N from the ceria lattice, thus serving as a direct fingerprint of the associative MvK-type pathway. Importantly, repeated pulsing behaviour further confirms this assignment: after a ¹⁵NH₃ pulse, the subsequent ¹⁴NH₃ pulse also yields m/z = 29, indicating that lattice nitrogen undergoes isotopic exchange (via formation and refilling of N vacancies), consistent with the proposed associative mechanism. After careful baseline correction and integration of the transient isotopic signals, the isotopic pulse MS analysis shows that, for the representative Ru/N-CeO₂ (650-6) catalyst at 450 °C, approximately 36.7 ± 1.5% of N₂ formation proceeds via the associative pathway, while 63.3 ± 1.5% arises from the dissociative pathway. These results demonstrate that both mechanisms operate concurrently under reaction conditions, with the associative pathway becoming particularly significant at lower Ru loadings where more N-doped sites remain accessible.

These findings and their mechanistic implications have been incorporated into the revised manuscript and Supplementary Information.

Revisions:

The following text has been added to the “Evaluation of Catalytic Performance” section on page 4 of the main manuscript:

“A series of catalysts with different Ru loadings and N contents were synthesised. The results showed that with an optimum N loading, a 2- to 3-fold reduction in the required Ru loading could be achieved (Fig. S7 and supplementary discussion 4).”

The following sentence was added to the “In-situ analysis and mechanistic understanding” section on page 10 of the main manuscript:

“By quantifying the area of the peaks corresponding to ¹⁴N-¹⁵N and ¹⁵N-¹⁵N during each ¹⁵NH₃ pulse, it is revealed that during the catalytic reaction, approximately 36.7 ± 1.5% of the products arises from the associative pathway, while 63.3 ± 1.5% originate from dissociative mechanisms.”

The following figure was added to the SI, page 8:

Fig. S7. Catalytic evaluation of catalysts with different Ru loading (wt.%) and N content (at.%) on ceria support. All measurements were conducted at 450 °C and $\text{WHSV} = 30,000 \text{ ml g}_{\text{cat}}^{-1} \text{ h}^{-1}$.

The revised discussion is copied below, which can be found in section “**Supplementary discussion 4: Effect of Ru loading versus N content towards the catalytic activity**” on page 35 of the Supporting Information:

“In general, the catalytic conversion of NH_3 increases with either higher N-doping levels or higher Ru loadings. For Ru loading, increasing the number of Ru nanoparticles leads to a larger population of Ru B_5 active sites, thereby enhancing the overall reaction rate. The effect of N doping is slightly different. As discussed in the main text, N dopants not only promote the intrinsic activity of Ru but can also serve as independent active sites that facilitate the associative reaction mechanism. Thus, increasing the N-dopant concentration also enhances catalytic activity.

As shown in Fig. S7, the NH_3 conversion achieved by Ru/N- CeO_2 (650-6) (Ru content: 0.3 wt.%; N content: 3.1 at.%) is ca. 38%, which is comparable to that of Ru/ CeO_2 (Ru content: 0.9 wt.%; N content: 0 at.%), ca. 35%. This demonstrates that introducing N dopants can reduce the required Ru loading by nearly threefold while maintaining similar catalytic performance.

It is also notable that the promotional effect of N dopants is more pronounced at lower Ru loadings. At low Ru loadings, a greater fraction of the support surface remains exposed,

allowing N-doped sites to contribute more significantly to the associative pathway. In contrast, at higher Ru loadings, increased coverage of the support surface by Ru nanoparticles decreases the number of accessible N active sites. As a result, the catalytic activity becomes dominated by the Ru B₅ sites, where the reaction primarily follows the dissociative mechanism, and the promotional influence of N dopants is diminished.”

Reviewer #3:

The manuscript describes a catalyst for ammonia decomposition with enhanced activity due to the use of a nitrogen-doped ceria in comparison to the non-doped support. Besides evaluating the catalytic performance, the mechanism is studied by operando NH_3 DRIFTS, isotopic exchange, machine-learning and DFT.

Apart from that, there are some issues regarding the experimental results and characterization and some of the conclusions regarding the mechanism proposed are hypothetical and not completely supported by the experimental characterization and literature. As it stands, I cannot recommend it for publication in Nature Communications. In the following are my comments/questions on some specific issues that should be addressed by the authors:

General response: We sincerely thank the reviewer for the careful evaluation of our manuscript and for the constructive comments provided. We appreciate the Reviewer's recognition of the experimental and computational breadth of the study, and we also thank him/her for highlighting several important points requiring further clarification. In response, we have carefully revised the manuscript to address each concern in detail, including improving the consistency of particle-size analysis, strengthening the discussion on the relative roles of Ru sites and N-doped sites, adding relevant literature on N-doping effects, and clarifying mechanistic statements. We have also incorporated additional results and discussions. We believe these revisions will significantly improve the quality and clarity of the manuscript. Point-by-point responses to all comments are provided below.

(1). In line 192, the average particle size is 2.2 nm. However, in supplementary information Figure S14 and EDS mapping, most of particles are larger, around 4-5 nm.

Response: We thank the Reviewer for highlighting this inconsistency in the original manuscript. The Ru nanoparticles indeed exhibit a relatively broad size distribution (from <1 to 5 nm), rather than a single uniform size, as shown in Figs. 2 and S15, which explains why some particles fall in the 4-5 nm range, as the Reviewer correctly pointed out. To present this more clearly, we have updated the Supporting Information with additional TEM images that capture a wider and more representative selection of Ru nanoparticles. We have included a particle size distribution analysis based on ~70 measured particles (revised Fig. S15). This analysis shows that, although larger particles are present, the overall distribution is centred at 2.1 ± 0.9 nm for Ru/N-CeO₂ (650-6) and 2.9 ± 0.7 nm for Ru/CeO₂. We hope this statistical analysis provides a clearer and more transparent description of the nanoparticle size distribution.

Revisions:

The following text has been adjusted and included in the "Effect of N doping on Ru" section on page 6 in the main manuscript:

"As illustrated in Fig. 2g and Fig. S15, Ru NPs exhibit a relatively broad size distribution (from <1 to 5 nm), rather than a single uniform size. Ru NPs are well dispersed across the surface of N-CeO₂ (650-6) with an average size of 2.1 ± 0.9 nm. The corresponding energy-dispersive X-

ray spectroscopy (EDS) mapping further demonstrates the homogeneous distributions of Ce, O, N and Ru on the catalyst (Fig. S16). The Ru NPs on pristine CeO₂ shows a slightly higher average particle size of 2.9 ± 0.7 nm (Fig. 2h) owing to the anchoring effect of N dopants as mentioned above.”

The following figure was added to the SI, page 16:

Fig. S15 TEM image and size distribution of Ru NPs on N-CeO₂ (650-6) support (a-d) and plain CeO₂ support (e-h).

(2). In line 155 it says: “the main contribution of catalytic activity arises from Ru”. This seems to be with strong contradiction with the larger part of the manuscript, which is devoted to the study of the associative mechanism induced by proximal N sites. Even the associative mechanism is mentioned in the title. To be more informative, it would help the quantification of the relative contribution to the activity of to each type of site, either Ru (dissociative mechanism) or N sites (associative mechanism).

Response: We thank the reviewer for raising this important point, and we apologise for the misleading wording in the original manuscript. Our original intention was to emphasise that the support alone shows negligible activity and that Ru remains indispensable for overall catalytic performance. However, we fully agree that the phrasing accidentally suggested a contradiction with our mechanistic findings. To avoid this confusion, we have revised the text to clarify that the dissociative pathway operates on Ru B₅ sites, whereas the associative pathway is initiated at N-sites but still requires the presence of Ru to facilitate N-H bond activation, as supported by our DFT results. This updated statement ensures consistency with the mechanistic insights established in the manuscript.

Regarding the quantification of the relative contributions from the associative (support-derived) and dissociative (Ru-derived) pathways, we used integrated peak areas from our isotopic pulse MS measurements to monitor the formation of ¹⁵N-¹⁵N (m/z = 30) and ¹⁵N-¹⁴N (m/z = 29).

Continuous $^{15}\text{NH}_3$ feeding, which would allow steady-state quantification, was not feasible due to the high cost and limited availability of $^{15}\text{NH}_3$. The pulsed-MS approach, however, provides reliable mechanistic insight because each isotopic pulse captures transient N_2 formation processes. Upon a $^{15}\text{NH}_3$ pulse, the $m/z = 30$ signal reflects N_2 generated solely through the dissociative pathway on Ru sites, whereas the $m/z = 29$ signal arises from coupling between ^{15}N from the feed and ^{14}N from the ceria lattice, thus serving as a direct fingerprint of the associative MvK-type pathway. Importantly, repeated pulsing behaviour further confirms this assignment: after a $^{15}\text{NH}_3$ pulse, the subsequent $^{14}\text{NH}_3$ pulse also yields $m/z = 29$, indicating that lattice nitrogen undergoes isotopic exchange (via formation and refilling of N vacancies), consistent with the proposed associative mechanism. After careful baseline correction and integration of the transient isotopic signals, the isotopic pulse MS analysis shows that, for the representative Ru/N-CeO₂ (650-6) catalyst at 450 °C, approximately $36.7 \pm 1.5\%$ of N_2 formation proceeds via the associative pathway, while $63.3 \pm 1.5\%$ arises from the dissociative pathway. These results demonstrate that both mechanisms operate concurrently under reaction conditions, with the associative pathway becoming particularly significant at lower Ru loadings where more N-doped sites remain accessible.

These results have been incorporated into the revised manuscript, and we believe they provide a clearer and more quantitative understanding of how both mechanistic pathways contribute to the overall catalytic activity.

Revisions:

The following text has been adjusted and can be found in the “Effect of N doping on Ru” section on page 5 in the main manuscript:

“Not surprisingly, the plain supports showed low activities, and even though the N-doped support exhibits a slightly higher activity than plain CeO₂, it is clear that the presence of Ru is essential for high catalytic performance. However, the origin of the observed activity remains to be clarified; specifically, the key question is whether the activity arises solely from Ru B₅ sites operating via the conventional dissociative mechanism, or whether synergistic interactions between Ru and N dopants also contribute through an associative pathway.”

The following sentence was added to the “In-situ analysis and mechanistic understanding” section on page 10 of the main manuscript:

“By quantifying the area of the peaks corresponding to ^{14}N - ^{15}N and ^{15}N - ^{15}N during each $^{15}\text{NH}_3$ pulse, it is revealed that during the catalytic reaction, approximately $36.7 \pm 1.5\%$ of the products arises from the associative pathway, while $63.3 \pm 1.5\%$ originate from dissociative mechanisms.”

(3). Some effects of N-doping of carbon materials are well known and widely reported in the literature but not sufficiently supported by a literature revision in this manuscript. For instance, these well-known effects are:

- “help to stabilise the Ru and inhibit further aggregation” (line 187)
- it is described a peak shift observed in the spectrum (in line 182)
- “Nitrogen doping also facilitates electron transfer...” (in line 133)

There are two references (31, 32) about N-doped carbon material characterization but they are not about ammonia decomposition. Relevant references about N-doping metal interaction (ACS Catal. 5 (2015) 2740-2753. <https://doi.org/10.1021/acscatal.5b00094>) and NH₃ decomposition are lacking (J. Phys. Chem. C 116 (2012) 26385–26395). The N-doping on a conductive support such as carbon can have different effects of N doping of a metal oxide which have lower electron conductivity. Therefore, references about N-doping of Ce would be also relevant.

Response: We thank the reviewer for this suggestion and for recommending additional relevant literature. We agree that the original manuscript did not sufficiently reference prior studies related to N-doping effects, particularly regarding their effects on ammonia decomposition. In the revised manuscript, we have incorporated the references recommended by the Reviewer, along with several additional studies, to more robustly support the claims regarding electron transfer, nanoparticle stabilisation, and spectral shifts induced by N-doping. These additional references provide a comprehensive literature review, supporting the observed effects of N-doping. The changes are shown below for this Reviewer’s convenience.

Revisions:

- “Nitrogen doping also facilitates electron transfer...” (original line 133).

We have added the following references: Carbon 48, 267–276 (2010) and J. Phys. Chem. C 116, 26385–26395 (2012), both of which investigate Ru on N-doped carbon supports for NH₃ decomposition. These studies clearly demonstrate enhanced electron donation from N dopants to supported metal nanoparticles, consistent with the effect described in our system.

- “Help to stabilise the Ru and inhibit further aggregation” (original line 187).

To support this point, we have added the following references: Chem. Eng. J. 156, 404–410 (2010) and ACS Catal. 7, 1197–1206 (2017). The first shows that N-doped carbon can suppress Ru sintering during NH₃ decomposition, while the second reports improved nanoparticle stability on N-doped TiO₂. Although studies specifically on N-doped ceria are limited, these examples from both conductive carbon supports and metal oxides provide complementary and relevant evidence for the stabilizing effect of N dopants.

- Peak shift in the spectrum (original line 182).

As suggested, we have added ChemCatChem 5, 3829–3834 (2013) and ACS Catal. 5, 2740–2753 (2015). The former reports the shifts in Ru oxidation state on N-doped carbon supports, while the latter provides a detailed discussion of electronic perturbation at metal-N interfaces, reinforcing our interpretation of XPS peak shifts.

(4). What is the actual Ru content for N-CeO₂ and CeO₂?

Response: The authors thank the reviewer for raising this important point. We realise that the actual Ru loadings were not emphasised clearly enough in the original manuscript. In the revised version, we now explicitly state the Ru contents (measured using ICP-MS) for all catalysts discussed (Table 1 and Table S2. We have also updated the relevant figures (Fig. 1b-f and Fig. S7) to clearly indicate the Ru loading associated with each catalyst. These revisions have made the relevant information available and have improved the clarity of the manuscript.

Revisions:

The following figure was added to the SI, page 8:

Fig. S7. Catalytic evaluation of catalysts with different Ru loading (wt.%) and N content (at.%) on ceria support. All measurements were conducted at 450 °C and WHSV = 30,000 ml g_{cat}⁻¹ h⁻¹.

The following table has been moved to the main manuscript (page 4-5):

Table 1 Catalytic activity comparison with Ru-based catalysts for ammonia decomposition.

Catalyst	Metal content /wt. %	Reaction Temp. / °C	WHSV /ml g _{cat} ⁻¹ h ⁻¹	NH ₃ Conv. /%	Apparent H ₂ formation rate /mmol g _{cat} ⁻¹ min ⁻¹	Reference
Ru/CNTs	5.0	450	30,000	43.7	14.6	27

Ru/CNFs	3.2	500	6,500	99.0	6.2	28
Ru/La_{0.33}Ce_{0.67}	1.8	450	6,000	99.9	6.7	29
Ru/CeO₂	1.0	350	22,000	ca. 32.0	8.1	30
Ru/c-MgO	4.7	450	30,000	80.6	26.5	31
Ru/MgO (111)	3.4	450	30,000	99.9	32.4	32
Ru/C12A7:e⁻	2.2	450	15,000	99.9	16.7	33
Ru/Ce₅/MgAl(600)	2.0	465	30,000	50.0	16.8	34
RuLaCs/Al₂O₃	1.0	450	5,000	99.0	5.2	35
Ru/CeO₂NR-v	0.5	450	15,000	ca. 80.0	13.3	36
Ru/CeO₂	1.6 ^[a]	450	30,000	43.2	14.5	This work
Ru/N-CeO₂ (650-6)	1.6 ^[a]	450	15,000	96.0	16.5	This work
Ru/N-CeO₂ (650-6)	1.6 ^[a]	450	30,000	79.5	26.6	This work
Ru/N-CeO₂ (650-6)	1.6 ^[a]	450	72,000	50.6	40.6	This work

Note: [a] Ru content of this work obtained using ICP-MS

The following Table has been updated in the SI section, page 29:

Table S2 Catalytic activity comparison with non-noble-metal based catalysts for ammonia decomposition.

Catalyst	Metal content /wt. %	Reaction Temp. /°C	WHSV /ml _{cat} ⁻¹ h ⁻¹	NH₃ Conv. /%	Reference
K-CoNi_{alloy}-MgO-CeO₂-SrO	60.0	450	12,000	87.5	²
CeO₂/Ni	60.0	500	30,000	72.4	³
Co@BaAl₂O₄	40.0	450	30,000	66.6	⁴
LaCoO₃/Co@NC/SBA-15	14.1	450	30,000	59.0	⁵
Mo₂N/SBA-15/ rGO	25.3	450	30,000	31.0	⁶
Fe-Co/MgO	74.0	500	7,200	ca. 47.0	⁷
Co-Ba/Y₂O₃	29.7	450	30,000	ca. 50.0	⁸
Ru/CeO₂	0.15 ^[a]	450	30,000	21.3	This work
Ru/CeO₂	0.31 ^[a]	450	30,000	28.6	This work
Ru/CeO₂	0.91 ^[a]	450	30,000	34.6	This work
Ru/CeO₂	3.23 ^[a]	450	30,000	58.0	This work
Ru/N-CeO₂ (650-2)	0.15 ^[a]	450	30,000	26.4	This work
Ru/N-CeO₂ (650-2)	0.31 ^[a]	450	30,000	30.9	This work
Ru/N-CeO₂ (650-2)	0.91 ^[a]	450	30,000	44.3	This work

Ru/N-CeO₂ (650-2)	3.23 ^[a]	450	30,000	61.2	This work
Ru/N-CeO₂ (650-4)	0.15 ^[a]	450	30,000	29.0	This work
Ru/N-CeO₂ (650-4)	0.31 ^[a]	450	30,000	33.3	This work
Ru/N-CeO₂ (650-4)	0.91 ^[a]	450	30,000	50.7	This work
Ru/N-CeO₂ (650-4)	3.23 ^[a]	450	30,000	76.2	This work
Ru/N-CeO₂ (650-6)	0.15 ^[a]	450	30,000	29.7	This work
Ru/N-CeO₂ (650-6)	0.31 ^[a]	450	30,000	38.1	This work
Ru/N-CeO₂ (650-6)	0.91 ^[a]	450	30,000	54.9	This work
Ru/N-CeO₂ (650-6)	3.23 ^[a]	450	30,000	82.7	This work

Note: [a] Ru content of this work obtained using ICP-MS.

(5). In line 163, the increase of reduction temperature along the N content is explained because N facilitates electron transfer. If facilitates electron transfer, it makes sense that the reduction should be favoured. In fact, Ru peak shift to more reduced species in the XPS spectrum.

Response: We thank the Reviewer for identifying this inconsistency and apologise for the confusion caused by the original wording. The H₂-TPR measurements were performed on fresh, unreduced samples immediately after wetness impregnation. Thus, the observed reduction peaks correspond to the temperature required to reduce oxidised Ru species (primarily RuO₂) to metallic Ru. On N-doped supports, the presence of nitrogen introduces stronger metal-support interactions, arising from coordination between N lone-pair electrons and cationic Ru species. This ionic interaction during the synthetic procedure stabilises the oxidised Ru phase and therefore shifts the reduction peak to higher temperatures. This effect reflects stronger binding of oxidised Ruⁿ⁺ species to the N-doped CeO₂ surface, rather than facilitated electron transfer.

By contrast, the XPS and XAS measurements were conducted on reduced catalysts, where Ru is already in the metallic state. In this case, the binding energy of Ru 3p shifts slightly due to the electron-donating properties of N dopants. The XPS Ru 3p_{3/2} peak appears at 461.4 eV for Ru/N-CeO₂ (650-6), compared with 462.0 eV for Ru/CeO₂, indicating slightly higher electron density on Ru when supported on N-CeO₂. This observation is consistent with the electron-transfer discussion in the main text.

To avoid ambiguity, the revised manuscript now discusses these two phenomena: TPR reduction behaviour and XPS electronic shifts, separately and more clearly, ensuring that their distinct interpretations are made clear to the reader.

Revisions:

The following text has been corrected and can be found in the “Effect of N doping on Ru” section on page 5 in the main manuscript:

“H₂-TPR on freshly prepared Ru supported on CeO₂ and N-CeO₂ materials has been carried out. It exhibits a clear peak in the range of 90-150 °C, attributed to the reduction of oxidised Ru species (Ruⁿ⁺) to metallic Ru⁰ (Fig. S11).⁴⁰ This peak shifts remarkably from 95 up to 144 °C for Ru on various nitrogen-doped CeO₂. During wetness impregnation, the lone pairs on surface-exposed nitrogen sites interact with the positively charged Ru species, resulting in stronger Ru metal support interaction (MSI). This enhanced MSI manifests as a shift of the Ru reduction peak to higher temperature and gives rise to the linear correlation observed between the surface nitrogen content and the reduction temperature of the Ru species (Fig. 2a).”

The following discussion has been added towards the “Effect of N doping on Ru” section on page 6 in the main manuscript:

“On another note, the XPS 3p_{3/2} spectrum of Ru⁰ shows a peak position deviation from 461.4 eV for Ru/N-CeO₂ (650-6) compared to 462.0 eV for the undoped counterpart. This suggests that for fully reduced samples, the electron density on Ru is still slightly higher for N-CeO₂ (650-6) compared to pristine support, which agrees with the electron transfer effect from N lone pair to the Ru metal as discussed above.”

(6). In line 48 it says “inverse correlation between the NH₃ binding energy...” It should be N₂ instead of NH₃ to be consistent also with the abstract line 23. Some reference could be added to support this.

Response: We thank the reviewer for identifying this inconsistency and apologise for the oversight. In the revised manuscript, we have corrected the terminology to “atomic N binding energy” for clarity and consistency with the abstract. In addition, we have included appropriate references [*J. Catal.* **328**, 36–42 (2015); *J. Catal.* **230**, 309–312 (2005).; *J. Catal.* **197**, 229–231 (2001)] to support this statement.

Revisions:

The following text has been corrected and can be found in the “Introduction” section on page 2 in the main manuscript:

“The barriers of these processes are constrained by the Brønsted-Evans-Polanyi (BEP) scaling relationship, which dictates an inverse correlation between the atomic N binding energy and the N-H bond activation barrier.⁴⁻⁶ During the reaction, stronger interaction between the atomic N from the reactant and a metal M active site would facilitate N-H bond dissociation due to the stronger back-donation of electrons from the metal sites to the anti-bonding orbital of the N-H bond.”

(7). It is not clear what N- proximal is. Is it equivalent to lattice N?. The N proximal site should be defined or depicted in a Figure like 4.

Response: We thank the reviewer for highlighting this point. In our terminology, N_{prox} refers to lattice nitrogen dopants located close to the Ru clusters on the CeO_2 surface, but not so close as to allow Ru-N bonding. The N_{prox} sites experience moderate metal-support interaction and, according to our analysis, exert the strongest promotional effect on the associative MvK pathway. They are distinct from (i) interfacial N dopants (N_{inter}), which are directly adjacent to Ru and strongly bound thereto, and (ii) distal N dopants (N_{dist}), which are far from any Ru cluster.

To clarify this definition, we have added a detailed computational analysis in the revised manuscript that explicitly correlates the Ru-N distance with the activation barrier of the associative mechanism. As shown in the revised Fig. S22, the rate-determining step (RDS) barrier exhibits a non-monotonic dependence: it decreases from the interfacial N site (Ru-N $\approx 1.7 \text{ \AA}$), reaches a minimum at the proximal N site ($\approx 4.1 \text{ \AA}$), and increases again for distal or Ru-free configurations. This trend identifies the proximal N configuration as the most active.

In addition, we have updated Fig. 4b in the main text to visually distinguish the three types of N dopants: N_{inter} , N_{prox} , and N_{dist} , thereby providing a clearer conceptual and visual representation consistent with the terminology used throughout the manuscript.

Revision:

The following text was added to the “Theoretical insights...” section, pages 10-11:

“As shown in Fig. S22, the calculated RDS barrier exhibits a non-monotonic dependence on the Ru- N_{doped} distance: it drops from the interfacial (Ru-N bond distance 1.7 \AA) to the proximal N site (Ru-N distance 4.1 \AA) but increases again for the distal and Ru-free CeO_2 cases. This non-monotonic trend identifies the proximal N configuration as the optimum, giving the lowest RDS barrier and thus the highest predicted activity.”

The edited Fig. 4b (page 12 in the revised manuscript) is shown below:

Fig. 4b. Schematic representation of the concerted reaction mechanisms taking place on the dual active sites on Ru/N-CeO₂ catalyst. The B₅ site on Ru, on which the dissociative pathway proceeds, is highlighted in grey. The N dopant active site, on which the associative pathway proceeds, is highlighted in red.

The following figure was added to the SI, page 23:

Fig. S22 Ru-N_{doped} distance-dependent modulation of the rate-determining step barrier of MvK type NH₃ decomposition on Ru/N-CeO₂.

(8). Consider moving Table S2 of the comparison into the main article. Moreover, since Ru is expensive, comparison with other less expensive transition metals would be also important.

Response: We thank the reviewer for this valuable suggestion. In the revised manuscript, we have incorporated additional comparisons with recently reported non-noble metal catalysts to address the importance of cost considerations for practical applications. We have also moved key performance data previously presented in Table S2 into the main text (now Table 1) and updated Table S2 in the SI to improve accessibility and readability. These additions strengthen the contextual comparison of our catalyst relative to both Ru-based systems and less expensive transition-metal alternatives.

Revisions:

The following table has been moved to the main manuscript (page 4-5):

Table 1 Catalytic activity comparison with Ru-based catalysts for ammonia decomposition.

Catalyst	Metal content /wt. %	Reaction Temp. / °C	WHSV /ml g _{cat} ⁻¹ h ⁻¹	NH ₃ Conv. /%	Apparent H ₂ formation rate /mmol g _{cat} ⁻¹ min ⁻¹	Reference
Ru/CNTs	5.0	450	30,000	43.7	14.6	27
Ru/CNFs	3.2	500	6,500	99.0	6.2	28
Ru/La_{0.33}Ce_{0.67}	1.8	450	6,000	99.9	6.7	29
Ru/CeO₂	1.0	350	22,000	ca. 32.0	8.1	30
Ru/c-MgO	4.7	450	30,000	80.6	26.5	31
Ru/MgO (111)	3.4	450	30,000	99.9	32.4	32
Ru/C12A7:e⁻	2.2	450	15,000	99.9	16.7	33
Ru/Ce₅/MgAl(600)	2.0	465	30,000	50.0	16.8	34
RuLaCs/Al₂O₃	1.0	450	5,000	99.0	5.2	35
Ru/CeO₂NR-v	0.5	450	15,000	ca. 80.0	13.3	36
Ru/CeO₂	1.6 ^[a]	450	30,000	43.2	14.5	This work
Ru/N-CeO₂ (650-6)	1.6 ^[a]	450	15,000	96.0	16.5	This work
Ru/N-CeO₂ (650-6)	1.6 ^[a]	450	30,000	79.5	26.6	This work
Ru/N-CeO₂ (650-6)	1.6 ^[a]	450	72,000	50.6	40.6	This work

Note: [a] Ru content of this work obtained using ICP-MS

The following Table has been updated in the SI section, page 29:

Table S2 Catalytic activity comparison with non noble metal based catalysts for ammonia decomposition.

Catalyst	Metal content /wt. %	Reaction Temp. /°C	WHSV /ml g_{cat}⁻¹ h⁻¹	NH₃ Conv. /%	Reference
K–CoNi_{alloy}–MgO–CeO₂–SrO	60.0	450	12,000	87.5	²
CeO₂/Ni	60.0	500	30,000	72.4	³
Co@BaAl₂O₄	40.0	450	30,000	66.6	⁴
LaCoO₃/Co@NC/SBA-15	14.1	450	30,000	59.0	⁵
Mo₂N/SBA-15/rGO	25.3	450	30,000	31.0	⁶
Fe-Co/MgO	74.0	500	7,200	ca. 47.0	⁷
Co-Ba/Y₂O₃	29.7	450	30,000	ca. 50.0	⁸
Ru/CeO₂	0.15 ^[a]	450	30,000	21.3	This work
Ru/CeO₂	0.31 ^[a]	450	30,000	28.6	This work
Ru/CeO₂	0.91 ^[a]	450	30,000	34.6	This work
Ru/CeO₂	3.23 ^[a]	450	30,000	58.0	This work
Ru/N-CeO₂ (650-2)	0.15 ^[a]	450	30,000	26.4	This work
Ru/N-CeO₂ (650-2)	0.31 ^[a]	450	30,000	30.9	This work
Ru/N-CeO₂ (650-2)	0.91 ^[a]	450	30,000	44.3	This work
Ru/N-CeO₂ (650-2)	3.23 ^[a]	450	30,000	61.2	This work
Ru/N-CeO₂ (650-4)	0.15 ^[a]	450	30,000	29.0	This work
Ru/N-CeO₂ (650-4)	0.31 ^[a]	450	30,000	33.3	This work
Ru/N-CeO₂ (650-4)	0.91 ^[a]	450	30,000	50.7	This work
Ru/N-CeO₂ (650-4)	3.23 ^[a]	450	30,000	76.2	This work
Ru/N-CeO₂ (650-6)	0.15 ^[a]	450	30,000	29.7	This work
Ru/N-CeO₂ (650-6)	0.31 ^[a]	450	30,000	38.1	This work
Ru/N-CeO₂ (650-6)	0.91 ^[a]	450	30,000	54.9	This work
Ru/N-CeO₂ (650-6)	3.23 ^[a]	450	30,000	82.7	This work

Note: [a] Ru content of this work obtained using ICP-MS.

Review for the Manuscript#: NCOMMS-25-74873-T

This manuscript presents an exceptional study on the role of nitrogen dopants as independent active sites for ammonia decomposition. The work is original, demonstrating convincingly that N-dopants can facilitate an associative Mars-van Krevelen (MvK) mechanism, thereby circumventing the limitations imposed by classical Brønsted-Evans-Polanyi (BEP) scaling relationships. The combination of experimental techniques, such as the isotopic labelling studies, with advanced theoretical calculations (MACE and DFT) provides evidence. This work is worthy publication if the authors can address the following points to further enhance its clarity and impact.

1. The manuscript excellently describes the creation of N-vacancies (V_N) during the MvK cycle but leaves the regeneration mechanism somewhat implicit. The description of N-sites as "redox-active" is central to the analogy with enzymes and ceria's oxygen redox chemistry, but the specific redox couple (e.g., $N^{3-} \leftrightarrow N_2$) and the re-filling of V_N could be elaborated.
2. The "proximal N-site" is a key concept for the highest activity, but its definition ("located near but not directly bonded to Ru") remains qualitative. Readers may wonder about the experimental evidence for its population and how it can be engineered.
3. The coexistence of the dissociative (on Ru) and associative (on N) pathways is well-established. However, a natural question is their relative contribution to the overall superior activity.
4. Figure 3g: Clearly annotating on the figure which pulses are $^{14}NH_3$ and which are $^{15}NH_3$ would improve clarity.
5. **Figure 4b:** The schematic is helpful but could be enhanced. Consider using different colored arrows or labels to distinctly differentiate the dissociative pathway on the Ru B5 site from the associative MvK pathway on the N-site. This would immediately visually reinforce the dual-pathway concept.
6. In the computational model section, it is highly suggested to conduct the MD simulations to make sure whether the constructed cluster are structurally stable and reasonable under the reaction conditions. This will directly impact the subsequent mechanistic study and the conclusion.